

# Presentation, Calibration and Testing of the DCESS II Earth System Model of Intermediate Complexity (version 1.0)

Esteban Fernández[1], Gary Shaffer[2]

[1]Graduate Program in Oceanography, Department of Oceanography, Faculty of Natural Sciences and Oceanography, University of Concepción, P.O. Box 160-C, Concepción, Chile
[2]Niels Bohr Institute, University of Copenhagen, 2100 Copenhagen Ø, Denmark

*Correspondence to*: Esteban Fernández (estfernandez@udec.cl)

**Abstract.** A new, Earth System Model of Intermediate Complexity, DCESS II, is presented that builds upon, improves and extends the Danish Center for Earth System Science (DCESS) Earth System model (DCESS I). DCESS II has considerably greater spatial resolution than DCESS I while retaining the fine, 100 m vertical resolution in the ocean. It contains modules for the atmosphere, ocean, ocean sediment, land biosphere and lithosphere and is designed to deal with global change simulations on scales of years to millions of years while using limited computational resources. Tracers of the atmospheric module are temperature, nitrous oxide, methane ($^{12,13}$C isotopes), carbon dioxide ($^{12,13,14}$C isotopes) and atmospheric oxygen. For the ocean module, tracers are conservative temperature, absolute salinity, water $^{18}$O, phosphate, dissolved inorganic carbon ($^{12,13,14}$C isotopes), alkalinity and dissolved oxygen. Furthermore, the ocean module considers simplified dynamical schemes for large-scale meridional circulation and sea-ice dynamics, stratification-dependent vertical diffusion, a gravity current approach to the formation of Antarctic Bottom Water and improvements in ocean biogeochemistry. DCESS II has two hemispheres with six zonal-averaged atmospheric boxes and twelve ocean boxes distributed across the Indian-Pacific, the Atlantic, the Arctic and the Southern Oceans. A new, extended land biosphere scheme is implemented that considers three different vegetation types whereby net primary production depends on sunlight and atmospheric carbon dioxide. The ocean sediment and lithosphere model formulations are adopted from DCESS I but now applied to the multiple ocean and land regions of the new model.

A model calibration was carried out for the pre-industrial climate and model steady-state solutions were compared against available modern-day observations. For the most part, calibration results agree well with observed data, included excellent agreement with ocean carbon species. This serves to demonstrate model utility for dealing with the global carbon cycle. Finally, two idealized experiments were carried out in order to explore model performance. First, we forced the model by varying Ekman transport out of the model Southern Ocean, mimicking the effect of Southern Hemisphere westerly wind variations and second, we imposed freshwater melting pulses from the Antarctic ice sheet on to the model Southern Ocean shelf. Changes in ocean circulation and in the global carbon cycle found in these experiments are reasonable and agree with results for much





more complex models. Thus, we find DCESS II to be a useful and computational-friendly tool for simulations of past climates as well as for future Earth System projections.

## 1 Introduction

The carbon cycle is the backbone of the Earth's climate system since acts as a main regulator of global mean atmospheric temperature via atmospheric concentration of carbon dioxide ($p\text{CO}_2$). This cycle may be considered to be composed of two domains. One domain is a fast one with large exchange fluxes and relatively "rapid" reservoir turnovers (from years to thousands of years). This domain encompasses carbon in the atmosphere, ocean, superficial (bioturbated) ocean sediments and on land in vegetation, soil and freshwater. A second, slower domain (from hundreds of thousands to millions of years) consists of huge carbon stores in rocks and sediments that exchange carbon with the fast domain through volcanic emissions of $\text{CO}_2$, chemical weathering, erosion and sediment formation on the sea floor. How this carbon is partitioned between the different Earth's reservoirs is what sets the $p\text{CO}_2$ in the atmosphere.

Earth system models (ESM) include one or both of these domains. They are thereby useful tools that can help us gain understanding of past climates as well as make future climate projections. Depending on their complexity and spatial resolution, ESMs can take days, weeks or even months to run model simulations on the range of time scales mentioned above while using large computational resources. The Danish Center for Earth System Science (DCESS) model (DCESS I, Shaffer et al., 2008) is a low-order ESM with a simple geometry and ocean physics that deals with both domains of the global-scale carbon cycle and is thereby suitable for investigating Earth System changes on scales of years to millions of years while taking only minutes to days to run. The DCESS I model has proven to be useful tool in such studies as documented by many stand-alone or intercomparison study publications (e.g. Eby et al., 2013; Harper et al., 2020; Joos et al., 2013; Macdougall et al., 2020; Shaffer, 2010; Shaffer and Lambert, 2018; Shaffer et al., 2009; Zickfeld et al., 2013). Here we present a new, Earth System Model, DCESS II, that contains great improvements in model geometry and physical/biogeochemical processes while retaining the simplicity and the spirit of the DCESS I model. Thus, DCESS II is a simple, fast and highly flexible ESM of intermediate complexity well suited to run long-term experiments in a relatively simple way with no need for large computational resources.

This paper is organized as follows: in Sect. 2 we describe the modules of atmosphere, ocean, ocean sediment, land biosphere and lithosphere. In Sect. 3, we present the model solution and calibration procedure and show results for the model steady state, pre-industrial simulation. In addition, we carry out two idealized experiments in order to explore and test model performance. Finally in Sect. 4 we discuss our results, outline future perspectives and present conclusions.



## 2 Model description

DCESS II is an intermediate complexity Earth System model containing atmosphere, ocean, land biosphere, lithosphere, and ocean sediment modules designed to deal with global climate simulations on scales of years to millions of years. It is an enhancement and extension of the original DCESS I model (Shaffer et al., 2008) and includes, for example, much improved horizontal resolution, simplified ocean dynamics and seasonal cycles. Model geometry consists of two hemispheres with a land/ocean area distribution and ocean depth distribution as shown in Fig. 1.

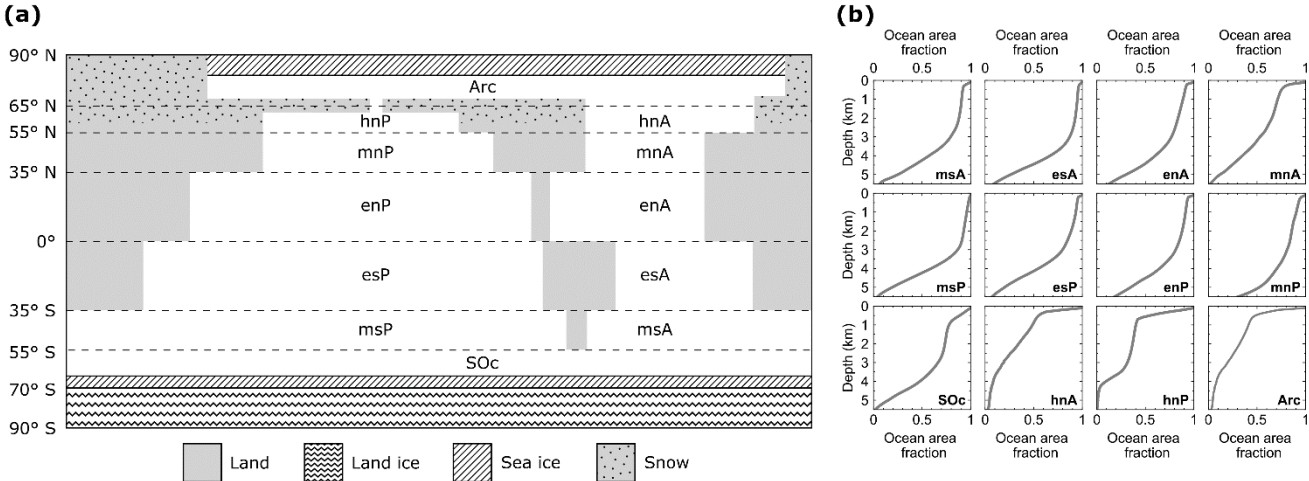

**Figure 1. (a)** Pre-industrial model land continental distribution and meridional boundaries of ocean boxes (dashed lines). There are twelve ocean boxes with their respective identifiers: Arc: Arctic Ocean, SOc: Southern Ocean, hn: high north, mn: mid north and en: equatorial north for Atlantic (A) and Indian-Pacific (P) Ocean. Box names in the Southern Hemisphere follow the same convention. High latitude North Pacific sector (hnP) and the Arctic Ocean are connected through the Bering Strait. Land south of 70° S is fully ice covered. Meridional sea-ice and snowline extents vary freely. **(b)** Observed present-day ocean area fraction profile of each ocean model box calculated from Amante and Eakins (2009).

The atmospheric module considers three zonally averaged boxes per hemisphere as low, mid, and high latitude sectors divided at 35° ($\phi_{35}$) and 55° ($\phi_{55}$) North/South latitudes (Fig. 2a). In the ocean module we include an extra division at 65° N which allows us to consider four global ocean basins (Atlantic, Indian-Pacific, Arctic, and Southern Ocean; for simplicity, hereafter Indian-Pacific Ocean will be called only Pacific Ocean). The Southern Ocean is 360° wide and extends up to 69° S where it interacts with an ocean shelf extending to 70° S. In all there are 12 ocean sectors (compared to only 2 in DCESS I). Each ocean sector is divided into 55 vertical layers with 100 m vertical resolution reaching 5500 m depth. An ocean sediment segment is assigned to each of the layers (Fig. 2b). For each sector, ocean layer and ocean sediment areas are determined from observed ocean depth distributions (Fig. 1b).





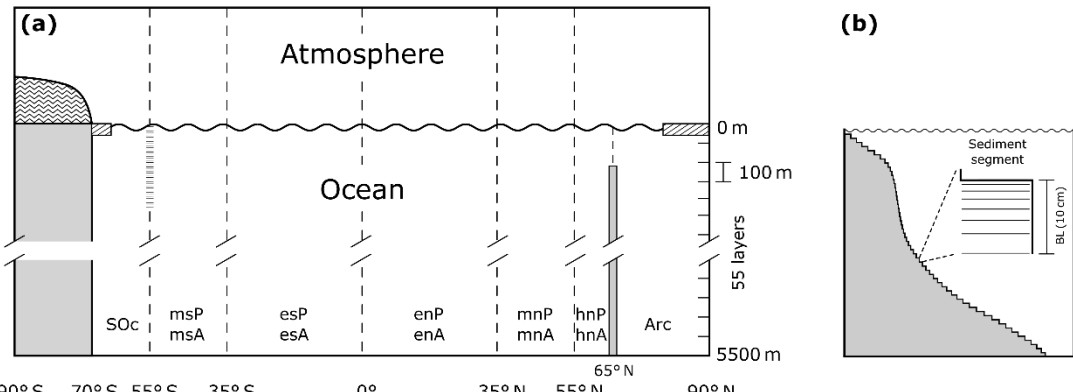

**Figure 2. (a)** Ocean-atmosphere cross-section view depicting meridional distributions of ocean and atmospheric boxes. There are six zonal-averaged atmospheric boxes and twelve ocean boxes (see also Fig. 1a). The shaded vertical bar at 55° S extending from the surface to 2000 m depth is a virtual barrier against meridional geostrophic flow above the Drake Passage sill depth. The vertical barrier at 65° N extending from the bottom up to 1000 m depth is a physical barrier associated with Denmark Strait bathymetry. **(b)** Sketch of an idealized vertical area profile for an ocean box (the actual profile for each box is based on bathymetry observations; see Fig. 1b). Also shown is an idealized sediment segment contained in a 100 m thick box layer (the area of each specific segment is again based on bathymetric observations). The bioturbated layer (BL) is assumed to be 10 cm thick and is divided into seven sublayers as shown.

The land biosphere module considers three different types of vegetation per hemisphere whose latitudinal limits varying dynamically according to climate conditions. All model modules are described in detail in the following sections.

### 2.1 Atmosphere exchange, heat balance and ice/snow extent

We use a simple, zonally integrated energy balance model for the near surface atmospheric temperature, $T_a$ (°C), forced by seasonally varying insolation, air-sea exchange, and meridional heat transport. In combination with simple sea ice and snow parameterizations, the model includes the ice-snow albedo feedback and the insulating effect of sea ice. Prognostic equations for mean $T_a$ in the 0°-35°, 35°-55° and 55°-90° zones ($T_a^{l,m,h}$) are obtained by integrating the surface energy balance over the zones as,

$$\Lambda \frac{dT_a^{l,m,h}}{dt} = r^2 \int_0^{2\pi} \int_{0,\phi_{35},\phi_{55}}^{\phi_{35},\phi_{55},\pi/2} (F^{\text{toa}} - F^T) \cos\phi \, d\phi d\xi \pm F^{\text{merid}} \tag{1}$$

where $\Lambda = A^{l,m,h} \rho_0 C_p b^{l,m,h}$, with $A^{l,m,h}$ the atmospheric box areas, $\rho_0$ the reference density of water, $C_p$ is the specific heat capacity and $b^{l,m,h}$ are the thicknesses chosen to yield observed seasonal cycles of $T_a^{l,m,h}$ (partially based on Olsen et al., (2005)), $r$, $\phi$ and $\xi$ are the Earth's radius, latitude and longitude respectively. Furthermore, $F^{\text{toa}}$ and $F^T$ are the vertical fluxes of heat through the top of atmosphere and the ocean surface, while $F^{\text{merid}}$ correspond to the loss (equatorward box) or gain




(poleward box) of heat due to meridional transport across $\phi_{35}$ or $\phi_{55}$. A no flux boundary condition has been applied at the poles. Meridional variations of $T_a$ in the model are represented by a fourth-order Legendre polynomial in sine of latitude,

$$T_a(\phi) = \sum T_n P_n (\sin \phi); \qquad n = 0, 2, 4 \tag{2}$$


where coefficients $T_0$, $T_2$ and $T_4$ are determined by matching the area-weighted, zone mean values of $T_a(\phi)$ to the prognostic mean sector values of $T_a^{l,m,h}$ in each hemisphere. Observations show that eddy heat fluxes in the mid-latitude atmosphere are much greater than advective heat fluxes there (Oort and Peixoto, 1983). By neglecting the advective heat fluxes, Wang et al., (1999) developed suitable expressions for $F^{\mathrm{merid}}$ and the associated moisture flux, $E^{\mathrm{merid}}$, in terms of $T_a$ and $\partial T_a / \partial \phi$ at $\phi$ (for

clarity we omit the index referencing low-mid and mid-high latitude boundaries, $\phi_{35}$ and $\phi_{55}$ respectively),

$$F^{\mathrm{merid}} = -\left[K_t + L_v K_q \exp\left(-5420 T_a^{-1}\right)\right] |\partial T_a / \partial \phi|^{m-1} \partial T_a / \partial \phi \tag{3}$$

$$E^{\mathrm{merid}} = -K_q \exp\left(-5420 T_a^{-1}\right) |\partial T_a / \partial \phi|^{m-1} \partial T_a / \partial \phi \tag{4}$$

where $K_t$ is a sensible heat exchange coefficient, $K_q$ is a latent heat exchange coefficient and $L_v$ is the latent heat of condensation ($2.25 \times 10^9$ J m$^{-3}$). From observations, $m$ is found to vary with latitude (Stone and Miller, 1980) and on this basis we take $m$ to be 2.5 and 1.7 at $\phi_{35}$ and $\phi_{55}$ respectively. The temperatures and temperature gradients entering in Eqs. (3)-(4) above are obtained from Eq. (2). The poleward freshwater flux crossing a latitude $\phi$ is evaporated from an equatorward ocean box and drains into the respective poleward Atlantic, Pacific, Arctic or Southern Ocean basin/box in accordance with observed

land catchment areas (Rodriguez et al., 2011). In this way, each ocean box receives a net freshwater flux from precipitation minus evaporation. The Arctic Ocean is an exception to this in as much as a fraction of $E^{\mathrm{merid}}$ crossing 55° N also drains there, emulating large northward flowing rivers in Canada and Siberia. Furthermore, an atmospheric zonal transport of freshwater of 0.15 Sv (1 Sv = $10^6$ m$^3$ s$^{-1}$) is prescribed from the tropical North Atlantic sector to the tropical North Pacific sector, emulating the effects of trade winds there (Dey and Döös, 2020; Lohmann and Lorenz, 2000). All these fluxes force respective ocean

sectors (see Sect. 2.3).

Based on a calibration procedure that makes use of monthly observed atmospheric temperature, we derived the following expression for interhemispheric heat transport at the equator,

$$F^{eq} = K_t^{eq} \partial T_a / \partial \phi + 1.1 + \left[K_q^{eq} L_v \partial T_a / \partial \phi - 1.1\right] \exp\left[-5420\left(\overline{T_a^{eq}}^{-1} - \overline{T_{a,PI}^{eq}}^{-1}\right)\right] \tag{5}$$



where $K_t^{eq}$ and $K_q^{eq}$ are sensible and latent heat coefficients and $\overline{T_a^{eq}}$ and $\overline{T_{a,PI}^{eq}}$ are the model and observed pre-industrial atmospheric mean temperature at the equator. For the $\overline{T_a^{eq}}$ calculation, Eq. (2) is used and evaluated at 0° latitude for Northern and Southern Hemispheres. The heat flux at the top of the atmosphere is taken as the balance between shortwave and longwave

radiation as follows,

$$F^{\text{toa}} = (1 - \alpha)Q - (A + BT_a) \tag{6}$$

where $\alpha$ is the planetary albedo, equal to 0.62 for ice and snow-covered areas and equal to $\alpha_0 - \alpha_2 (3 \sin^2 \phi - 1)$ otherwise.

This formulation includes the effect of mean cloud cover and lower solar inclination at higher latitudes (Hartmann, 2016), and $Q$ is the orbital, seasonal, and latitudinal (and if needed, paleo-time)-varying short-wave radiation (Berger and Loutre, 1991). The albedo of non-snow/ice-covered areas varies with vegetation type since forested areas have lower albedo than non-forested areas (Bonan, 2008). We adopt the approach in Eichinger et al., (2017) that include this effect by relating the $\alpha_0$ factor to the vegetation type such as $\alpha_0 = 0.3 - \gamma (1 - f(\delta T_a)/f_0)$ where the factor 0.3 is the present-day value of $\alpha_0$, and $\gamma$ is a multiplier

equal to 0.02. $f(\delta T_a)$ is the ratio between the area of Grasslands/Savanna/Deserts zone and the total non-snow/ice covered area as a function of $\delta T_a$, the deviation of atmospheric temperature from the present-day value ($f_0$ is the ratio at $\delta T_a = 0$; see Sect. 2.5 for details about vegetation zones). Finally, $\alpha_2$ is equal to 0.0875.

The second term on the right-hand side of Eq. (6) is the outgoing longwave radiation (Budyko, 1969), whereby $A$ and $BT_a$ are

the flux at $T_a = 0$ and the deviation from this flux respectively. This simple formulation includes implicitly the radiative effects of changes in cloud cover and in atmospheric water vapor content. Greenhouse gas forcing is modelled by taking $A$ to depend on deviations of (prognostic) atmospheric partial pressures of carbon dioxide, methane, and nitrous oxide ($p$CO$_2$, $p$CH$_4$ and $p$N$_2$O) from their pre-industrial values (PI), such that

$$A = A_{\text{PI}} - \left( A_{\text{CO}_2} + A_{\text{CH}_4} + A_{\text{N}_2\text{O}} \right) \tag{7}$$

where expressions for $A_{\text{CO}_2}$, $A_{\text{CH}_4}$ and $A_{\text{N}_2\text{O}}$, as taken from Byrne and Goldblatt (2014), are well suited for low and high atmospheric greenhouse gas concentrations. Our pre-industrial baseline values of $p$CO$_{2,\text{PI}}$, $p$CH$_{4,\text{PI}}$ and $p$N$_2$O$_{\text{PI}}$ are 280, 0.72 and 0.27 $\mu$atm respectively. For simplicity and for the consideration of possible paleo-applications, we take constant values of

$A_{\text{PI}}$ and B for all atmospheric boxes (see Table 1). For each ocean sector, air-sea heat exchange is calculated according to Haney (1971) as,

$$F^T = -L_o - \lambda \left( T_a^{if} - T_o \right) \tag{8}$$





where $L_o$ is the direct (solar) heating of the ocean surface layer, taken to be 40, 20, and 0 W m$^{-2}$ for the low, mid and high latitude sectors respectively, $\lambda$ is a constant bulk transfer coefficient, as a good approximation taken to be 30 W m$^{-2}$ °C$^{-1}$ but set to zero for areas covered by sea ice (Haney, 1971), $T_a^{if}$ is the ice-free mean atmospheric temperature for each sector, and $T_o$ is the zone mean, ocean surface temperature of each ocean box.

**Table 1. Atmosphere module parameters.**

| Parameter | Symbol | Value |
|---|---|---|
| Sensible heat exchange coefficient | $K_t$ $(\phi_{eq}/\phi_{lm}/\phi_{mh})$ | $2.40\times10^{15}/4.21\times10^{11}/3.30\times10^{12}$ J s$^{-1}$ °C$^{-2.5}$ |
| Latent heat exchange coefficient | $K_q$ $(\phi_{eq}/\phi_{lm}/\phi_{mh})$ | $-7.11\times10^5/1.56\times10^{10}/2.29\times10^{11}$ m$^3$ s$^{-1}$ °C$^{-2.5}$ |
| Pre-industrial, zero-degree, longwave radiation | $A_{PI}$ | 206.58 W m$^{-2}$ |
| Temperature sensitivity of longwave radiation | B | 2.21 W m$^{-2}$ °C$^{-1}$ |
| Inverse timescale for sea-ice advance and retreat | Γ | $3\times10^{-2}$ s$^{-1}$ |
| Ice-water heat exchange coefficient | $\kappa_0$ | 150 W m$^{-2}$ °C$^{-1}$ |

### 2.1.1 Sea ice and snow cover

Sea ice plays a pivotal role in the Earth's climate system, influencing radiative balance and thus global atmospheric temperature through its impact on surface albedo and air-sea heat exchange processes. For the sea ice cover, we take the same simple

dynamical formulation as in Olsen et al., (2005) for the seasonal, equatorward sea ice extent. This formulation takes advantage of the meridional profile of $T_a$ and assumes that: a) sea ice advance is proportional to the inverse timescale of cooling ($\tau^{adv}$) of the ocean mixed layer to the freezing temperature ($T_f$) by heat loss to the atmosphere and b) that the retreat is taken to be proportional to the inverse timescale of melting ($\tau^{ret}$) of a seasonal sea ice cover. These inverse timescales are expressed as

$$\tau^{adv} = \frac{\lambda}{\rho_0 c_p d_u} \frac{T_a(\phi_i) - T_o}{T_o - T_f} \tag{9}$$

$$\tau^{ret} = \frac{k_i}{\delta_i^2 \rho_i L_i}[T_a(\phi_i) - T_f] + \frac{\kappa_0}{\delta_i \rho_i L_i}[T_o - T_f] \tag{10}$$

where $d_u$ is the mixed layer depth (100 m), $T_a(\phi_i)$ is the atmospheric temperature at the sea ice edge, $T_o$ is the ocean surface temperature, $k_i$, $\delta_i$, $\rho_i$, $L_i$ and $\kappa_0$ are the thermal conductivity of ice (2.0 W m$^{-2}$ °C$^{-1}$), the sea ice thickness (2 m), the density

of ice (917 kg m$^{-3}$), the latent heat of fusion of ice ($3.34\times10^5$ J kg$^{-1}$) and the heat transfer coefficient between ice and water (150 W m$^{-2}$ °C$^{-1}$; Bendtsen, 2002) respectively. Thus, changes in sea ice line position are determined individually for the Arctic/Northern Atlantic, Northern Pacific and Southern Ocean as,





$$\frac{\partial \phi_i}{\partial t} = -\Gamma(\tau^{adv} - \tau^{ret}) \tag{11}$$


where $\Gamma$ is a free parameter that together with $\kappa_0$ has been chosen to match the observed seasonal amplitude and the annual mean position of sea ice cover respectively. Land areas are covered by snow where $T_a(\phi) \leq 0$ °C and taken to respond instantaneously to atmospheric temperature changes. Furthermore, present-day Northern Hemisphere ice sheets are prescribed as a constant area with ice albedo throughout the year independent of snow line position. All of Antarctica is covered by ice

in the model.

## 2.2 Atmospheric chemistry and air-sea gas exchange

For each atmospheric box we consider partial pressures of $^{12,13,14}CO_2$, $^{12,13}CH_4$, $N_2O$ and $O_2$. The prognostic equation for the partial pressure of a gas $\chi$ is taken to be

$$\frac{dp(\chi)}{dt} = \frac{1}{v_a}\left[A_o^{if}\Psi_S(\chi) + \Psi_I(\chi) \pm \Psi_T(\chi)\right] \tag{12}$$

where $v_a$ is an atmospheric mole volume, $A_o^{if}$ an ice-free ocean surface area, $\Psi_S$ is an air-sea gas exchange flux, $\Psi_I$ are sources or sinks within the atmosphere or net transports to or from the atmosphere via weathering, volcanism, interaction with land biosphere and, for recent times, anthropogenic activities, and $\Psi_T$ is the gas transport between adjacent atmospheric boxes.


Air-sea exchange for $^{12}CO_2$ for each atmospheric box is written as (for simplicity we omit atmospheric box index superscript)

$$\Psi_S = k_w \eta_{CO_2}\left(pCO_2 - pCO_{2,w}\right) \tag{13}$$

where the gas transfer velocity $k_w$ is $0.39\, u^2(Sc/660)^{-0.5}$ with $u$ the long-term annual mean wind speed at 10 m above ocean surface, $Sc$ the $CO_2$ Schmidt number that depends on prognostic temperatures of the ocean surface layers (Wanninkhof, 1992), and $\eta_{CO_2}$ is the $CO_2$ solubility that is a function of temperature and salinity of the surface ocean layer (Weiss, 1974). $pCO_{2,w}$ is the prognostic $CO_2$ partial pressure at the ocean surface layer equal to $[CO_2]/\eta_{CO_2}$ whereby $[CO_2]$ is the prognostic dissolved $CO_2$ concentration of the ocean surface layer calculated from ocean carbonate carbon chemistry (see Sect. 2.4). Wind speeds

are not calculated in our simplified atmosphere module, so we use observed values from the NOAA/CIRES/DOE 20th Century Reanalysis (V3) dataset for each ocean sector. For extremely warm climates experiments we use the Schmidt number formulation from Gröger and Mikolajewicz (2011) who demonstrated that the Wanninkhof (1992) formulation underestimates $Sc$ in such conditions.





Air-sea exchange for $^i$CO$_2$ ($i$=13 and 14) is given by

$$\Psi_S^i = k_w \eta_{CO_2} \, ^i\alpha_k \left( ^i\alpha_{aw} p\,^i CO_2 - ^i\alpha_{wa} pCO_{2,w} \frac{[DI^iC]}{[DIC]} \right) \tag{14}$$

where $^i\alpha_k = 0.99912$ is the kinetic fractionation factor (Zhang et al., 1995), $[DI^iC]$ and $[DIC]$ are concentrations of dissolved

inorganic carbon in the surface ocean layer, $^i\alpha_{aw}$ is the fractionation factor due to different solubilities in the equilibration

process and $^i\alpha_{wa}$ is the fractionation factor in the dissociation reactions associated with ocean carbonate chemistry expressed

as,

$$^i\alpha_{wa} = \frac{[CO_2] + ^i\alpha_{HCO_3}[HCO_3^-] + ^i\alpha_{CO_3}[CO_3^{2-}]}{DIC} \tag{15}$$


where $^i\alpha_{HCO_3}$ and $^i\alpha_{CO_3}$ are individual fractionation factors for the carbon species $[HCO_3^-]$ and $[CO_3^{2-}]$ and surface layer values

for these species are obtained from ocean carbonate chemistry calculations. Moreover, $^i\alpha_{HCO_3}$ and $^i\alpha_{CO_3}$ are functions of

surface layer temperature (Zhang et al., 1995). As the rate at which different isotopes of a chemical element take part in a

chemical reaction depends on their mass, we assume the $^{14}$C/$^{12}$C fractionation to be twice as strong as for $^{13}$C/$^{12}$C, then $^{14}\alpha =$

$1 - 2(1 - ^{13}\alpha)$.

For air-sea exchange of oxygen for each ocean surface sector

$$\Psi_S = k_w \left( \eta_{O_2} pO_2 - [O_2] \right) \tag{16}$$


with $k_w$ as before but with O$_2$ Schmidt numbers that depend on prognostic temperatures of the ocean surface layers (Keeling

et al., 1998). The O$_2$ solubility ($\eta_{O_2}$) was converted from Bunsen solubility coefficients that depend on prognostic temperatures

and salinities (Weiss, 1970) to model units using the ideal gas mole volume, $[O_2]$ is the prognostic dissolved oxygen

concentration in the ocean surface layer (see Sect. 2.4). At present we do not include air-sea exchange of methane and nitrous

oxide in this model version. However, this could be readily accomplished upon need, as this has been successfully implemented

in the DCESS I model (Shaffer et al., 2017).

With regard to sources/sinks (the second term on the right-hand side of Eq. (12)), the model considers the following

sources/sinks for each atmospheric tracer:






For carbon dioxide, there is net exchange with the land biosphere, oxidation of atmospheric methane, volcanic input, weathering of "old" organic carbon in rocks and weathering of carbonate and silicate rocks. As needed, anthropogenic $CO_2$ sources may be added associated with fossil fuel burning and/or land use change. All these sources/sinks are also considered for atmospheric $^{13}CO_2$. For $^{14}CO_2$ the same sources/sinks as above are included, except for old (and thus $^{14}$C-free) carbon

sources which are inputs from volcanoes, organic carbon weathering and fossil fuel burning. In addition, $^{14}$C is produced naturally in the atmosphere via cosmic ray flux and, in recent times, by atomic bomb testing. For each atmospheric box, the cosmic ray source of $^{14}CO_2$ can be expressed as $A^{l,m,h} P_{14C}^{l,m,h}/A_{vg}$, where $A_{vg}$ is the Avogadro number and $P_{14C}^{l,m,h}$ (in atoms m$^{-2}$ s$^{-1}$) is the magnitude of $^{14}$C production, chosen here to match the estimated pre-industrial atmospheric $^{14}$C concentration such that $\Delta^{14}C_{atm} \sim 0$ ‰. A small amount of atmospheric $^{14}$C enters the land biosphere and decays there radioactively. A smaller

fraction decays directly in the atmosphere becoming an atmospheric sink of $^{14}CO_2$ with a decay rate $\lambda_{14C}$ of $3.84\times10^{-12}$ s$^{-1}$. By far, most of the $^{14}$C produced enters the ocean by air-sea exchange and by far most of this isotope decays within the ocean. Finally, a small amount of $^{14}$C enters the ocean sediment via sinking of biogenic particles and part of this returns to the ocean due to remineralization/dissolution in the ocean sediment.

For methane there is production within the land biosphere (see Sect. 2.5) and consumption associated with OH radicals in the troposphere. Since this reaction depletes the concentration of these radicals, atmospheric lifetime of $CH_4$ grows as methane concentration rises. We include this effect in the model by taking the atmospheric methane sink to be $\lambda_{CH_4}\, pCH_4$, with $\lambda_{CH_4} = M(pCH_4)/\tau_{CH_4,PI}$ whereby $M(pCH_4)$ is a fitted function from several modelling studies that consider a wide range of $pCH_4$ values and $\tau_{CH_4,PI}$ is the atmospheric pre-industrial lifetime of methane equal to 9.5 years (Shaffer et al., 2017). Potential

sources like melting of methane hydrate in the arctic tundra and in ocean sediments and by human activities may be included in the model as needed. For $^{13}CH_4$ the same processes are considered but with their respective fractionation factors (Sect. 2.5) and adding the $p^{13}CH_4/p^{12}CH_4$ ratio to the above OH-related atmospheric methane sink formulation.

For nitrous oxide there is production within the land biosphere and consumption in the atmosphere, here mainly due to

photodissociation in the stratosphere. This is modelled as $\lambda_{N_2O} pN_2O$, with $\lambda_{N_2O} = v_a/\tau_{N_2O}$ where $\tau_{N_2O}$, the atmospheric lifetime of $N_2O$, is taken to be 150 years. Potential sources like $N_2O$ flux from ocean denitrification and by human activities may be included in the model as needed.

For oxygen, there is consumption associated with oxidation of atmospheric methane and reaction with OH radicals (see above),

and sink (source) from organic matter remineralization (photosynthesis) on land. Furthermore, there are long-term atmospheric oxygen sinks due to weathering of organic carbon in rocks and oxidation of reduced carbon emitted in lithosphere outgassing. A long-term steady state of $pO_2$ is achieved in the model when these latter sinks balance net $O_2$ outgassing from the ocean from less $O_2$ consumption than production there due to burial of organic matter in the model ocean sediments. Additional sinks





(sources) of atmospheric $O_2$ associated with recent land use change and with burning of fossil fuels may be included in the
model as needed.

The last term in Eq. (12) representing the gas transport between two adjacent atmospheric boxes is modelled as

$$\Psi_T(\chi) = -k_\chi \frac{dp(\chi)}{d\phi} \tag{17}$$


with $k_\chi = 6\times10^{-8}$ s$^{-1}$ based on data-model comparisons we made using the DCESS I model (Shaffer et al., 2008). For
simplicity, we take the same value for all gases and for all atmospheric boundaries.

Evaporation and precipitation modify both salinity and oxygen isotopic composition in the surface ocean. Fractionation due to
evaporation enriches (depletes) the $^{18}O$ content in sea water (water vapor) from net evaporative model ocean (atmospheric)
zones. Further depletion in water vapor takes place due to fractionation in the condensation process as the air mass cools in its
poleward path. Given the importance of $^{18}O$ in paleoclimate studies, we incorporate atmospheric cycling of oxygen isotopes
of water following Olsen et al., (2005), which gives the $^{18}O$ content in the well-known delta notation ($\delta^{18}O_w$) relative to
Standard Mean Ocean Water (SMOW). This approach takes advantage of meridional atmospheric temperature profiles
estimating $\delta^{18}O_a$ at every atmospheric box division, where subindex "$a$" refers to isotopic excursion of $^{18}O$ in the atmosphere.

### 2.3 Ocean circulation and mixing

Simplified ocean dynamics in the model consist of a balance between the pressure gradient force and linear (Rayleigh) friction
acting on the meridional velocity. Model flow is defined by this relation and hydrostatic and continuity equations,

$$-\frac{1}{r\rho_0}\frac{\partial P}{\partial \phi} - r_f v = 0 \tag{18}$$

$$-\frac{1}{\rho_0}\frac{\partial P}{\partial z} - \frac{\rho}{\rho_0} g = 0 \tag{19}$$

$$\frac{1}{r\cos\phi}\frac{\partial v \cos\phi}{\partial \phi} + \frac{\partial w}{\partial z} = 0 \tag{20}$$

where $r_f$ is a friction coefficient (Table 2), $v$ and $w$ are the meridional and vertical velocity components respectively, $P$ is the
pressure, $g$ is gravity and $\rho$ is the water density, calculated using the non-linear function of temperature, salinity, and pressure
according to the TEOS-10 standard (International Thermodynamic Equation Of Seawater, IOC et al., (2010)) but modified for
Boussinesq ocean models (Roquet et al., 2015).





**Table 2. Ocean module physical and biogeochemical parameters as well as applied Redfield ratios.**

| Parameter | Symbol | Value |
|---|---|---|
| Ocean Physics | | |
| Meridional flow friction coefficient | $r_f$ | $1\times10^{-4}$ s$^{-1}$ |
| Deep ocean horizontal diffusivity | $K_h^d$ | $1\times10^{3}$ m$^2$ s$^{-1}$ |
| Surface ocean horizontal diffusivity | $K_h^s$ | $2.5\times10^{4}$ m$^2$ s$^{-1}$ |
| e-folding depth for horizontal diffusivity scale | $z_g$ | 200 m |
| Glacial melt water flux to Southern Ocean shelf | $F_g$ | 0.07 Sv |
| Sea ice export flux from Southern Ocean shelf | $F_i$ | 0.13 Sv |
| Ocean Biochemistry | | |
| Limitation factor (Southern Ocean/Arctic Ocean) | $L_f$ | 0.15/0.30 s$^{-1}$ |
| Upper limit for rain ratio | $r_{Cal,max}$ | 0.35 |
| Steepness factor for calcite production | $\mu$ | 0.16 |
| Nutrient remineralization length scale | $\xi_N$ | 833 m |
| Carbon remineralization length scale | $\xi_C$ | 769 m |
| Calcite dissolution length scale | $\xi_{Cal}$ | 2500 m |
| Redfield ratios | | |
| Carbon to phosphate | $r_{CP}$ | 106 |
| Alkalinity to phosphate | $r_{AlkP}$ | 16 |
| Oxygen to phosphate (nutrient component) | $r_{ONP}$ | 32 |
| Oxygen to phosphate (carbon component) | $r_{OCP}$ | 118 |

Since there are no meridional boundaries at the Drake Passage, a geostrophic flow cannot be maintained above the sill depth there. To account this feature in our simplified model, a virtual barrier against meridional flow is placed at 55° S from surface to 2000 m depth (Fig. 2). Equatorward Ekman transport there, forced by prevailing westerlies winds, is included by prescribing

a net northward volume flux ($E_k$) in the surface layer taken to be 30 Sv. This transport, which carries Southern Ocean surface water properties, is injected into the "South Atlantic/South Pacific" sector on the corresponding density level, forming the model's Antarctic Intermediate Water (AAIW). Moreover, a northward surface flow of 1 Sv at 65° N is prescribed to consider the net ocean exchange between the North Pacific and the Arctic Ocean across the Bering Strait. This flow carrying surface North Pacific properties enters on the corresponding density level in the Arctic Ocean. At all other latitudes, meridional flow

is set to zero in the surface layer where exchanges rely entirely on horizontal and vertical mixing processes (Shaffer and Olsen, 2001). The zonally averaged velocity at each meridional zone boundary is related to the density field according to





$$v(z > d_u) = -\frac{g}{ar_f\rho_0}\int_0^z \frac{\partial\rho}{\partial\phi}dz - \frac{g}{ar_f}\frac{\partial\eta}{\partial\phi} \tag{21}$$

where $\eta$ is the sea level and $d_u$ is the depth of the mixed layer or, for Southern Ocean, the Drake Passage sill depth and for Arctic Ocean, the Denmark Strait sill depth. Sea levels of the model ocean zones are adjusted instantaneously to conserve mass and to form sea level gradients in Eq. (21) that lead to ocean meridional transports needed to balance the Ekman transport and atmospheric freshwater forcing at the ocean surface. Finally, the vertical flow is calculated from continuity equation, given meridional flow, Ekman transport, atmospheric freshwater forcing and shelf exchange (see below).


The effect of wind-driven gyre mixing is parameterized by a depth-dependent, horizontal diffusivity,

$$K_h(z) = K_h^d + K_h^s \exp\left(\frac{z}{z_g}\right) \tag{22}$$

where $K_h^d$ and $K_h^s$ denote the deep and surface horizontal diffusivities and $z_g$ is an e-folding, gyre depth (Table 2). At the Southern Ocean where wind-driven gyre mixing is not important, $K_h^s$ is decreased by 90 %. The same approach is taken at the North Pacific sector where North Pacific Intermediate Water (NPIW) is formed with a reduction of 50 % in $K_h^s$. With this setting, the model can reproduce global-scale intermediate water masses (AAIW and NPIW) in a simple way as reported by other model studies (England, 1993; Stocker et al., 1994). This is one of the few "regional tunings" in the model. The vertical

diffusivity, $K_v$, is modelled using a stratification-dependent form, such that

$$K_v(z) = \min\left[K_v^{\max}, K_v^0\left(\frac{N}{N_0}\right)^{-\gamma}\right]; \text{for } N > 0 \tag{23}$$

where $N$ is the Brunt-Väisälä frequency equal to $[(g/\rho_0)(\partial\rho/\partial z)]^{1/2}$. According to Gargett and Holloway (1984), we set
$\gamma = 0.5$ indicating a relatively weak dependence on stratification associated with diapycnal mixing via breaking of internal waves. Furthermore, $K_v^0$ and $N_0$ are the respective scale values of $K_v$ and $N$ taken as global representative thermocline values equal to $3\times10^{-5}$ m$^2$ s$^{-1}$ and $10^{-2}$ s$^{-1}$ respectively (Kunze, 2017). For unstable stratification $K_v$ is equated with $K_v^{\max}$ ($8\times10^{-4}$ m$^2$ s$^{-1}$), accounting for model convective adjustment.

### 2.3.1 Southern Ocean shelf and Antarctic Bottom Water formation

Antarctic Bottom Water (AABW) constitutes a major ocean water mass filling and ventilating the abyssal ocean on a global scale (Johnson, 2008; Orsi et al., 1999). Also, AABW plays a crucial role through the global overturning circulation in the transport and redistribution of heat, salt, nutrients and carbon (among other tracers) and thus affects atmospheric CO$_2$ concentrations and thereby global climate (Orsi, 2010; Purkey and Johnson, 2013).





AABW originates from dense shelf water, formed by air cooling and brine-rejection during sea ice formation, that can flow down the slope reaching abyssal ocean depths (Gordon, 2019). To address this process, we include a 500 m deep Southern Ocean shelf (Heywood et al., 2014) between 69° S and 70° S. Furthermore, we assume that there is enough air-sea heat exchanges on the shelf to maintain temperature there at freezing temperature $T_f$. Then, the prognostic equation for all shelf tracers but temperature ($\psi_{sh}$) is,


$$\frac{\partial \Psi_{sh}}{\partial t} = \frac{1}{V_{sh}}\left[\left(F_g - F_c\right)\Psi_g - F_i\Psi_i + F_r\Psi_{SO} - F_{sl,0}\Psi_{sh} + F_{ge}A_{sh}^{if}\right] \tag{24}$$

where $V_{sh}$ is the shelf volume, $F_g$ is the freshwater flux due to ice sheet/calved iceberg melt and precipitation and $F_C$ is the iceberg export from the shelf taken as 0.05 Sv and 0.02 Sv respectively. Furthermore, $F_i$ is the sea-ice export to the Southern

Ocean taken as 0.13 Sv (Haumann et al., 2016), $F_r$ is the replacing flux from the Southern Ocean to the shelf and $F_{sl,0}$ is the volume flux leaving the shelf at the shelf break (see below). Each of these fluxes carry their own tracer values ($\psi_g$, $\psi_i$, $\psi_{SO}$ and $\psi_{sh}$). Values of $\Psi_g$ are zero for salinity, phosphate and alkalinity tracers, $\eta_{CO_2}p^{12,13,14}CO_2$ for DI$^{12,13,14}$C, $\eta_{O_2}O_2$ for dissolved oxygen and a constant value of -40 ‰ for $\delta^{18}O_w$. The latter is a rough estimate for the mean present day $\delta^{18}O_w$ value of ice exported from Antarctica based on snow input values and ice sheet flow (Masson-Delmotte et al., 2008). For $\Psi_i$, sea-ice

takes the shelf water value, except for salinity which is taken as constant equal to 5 g kg$^{-1}$. Furthermore, a fraction of the freshwater flux crossing 55° S falls directly into the Southern Ocean and the rest goes to Antarctica. In steady-state the amount going to Antarctica is equal to $F_g$. The last right-hand term corresponds to the air-sea gas exchange where $F_{ge}$ is the gas flux and $A_{sh}^{if}$ is the ice-free surface shelf area, meant mainly to represent coastal polynyas. Here we take $A_{sh}^{if}$ to be 20 % of the total shelf area.


If it is denser than ambient Southern Ocean water, shelf water flows down along the continental slope. On a path to the deep ocean, downslope flow takes up ambient water via entrainment, typically increasing its volume more than twice its value at shelf break to finally fill the abyssal ocean as AABW (Orsi, 2002; Orsi et al., 2001). In order to include a comparable AABW formation process in our model, we address this using a simple formulation of entrainment following approaches described in

Baines (2005, 2008). The governing equations for the downslope flux ($F_{sl}$) and the plume height (or thickness, $H$) normal to the slope are

$$\frac{dF_{sl}}{ds} = E\frac{F_{sl}}{H} \tag{25}$$

$$\frac{dH}{ds} = 2E + C_d - S_2 Ri\tan\theta \tag{26}$$






where $s$ is the downslope distance from the shelf break depth obtained from observed bathymetry (Amante and Eakins, 2009) and $E$ is the entrainment coefficient equal to $E_0(1 - Ri/Ri_c)$ for $0 \leq Ri \leq Ri_c$ and zero otherwise. Here $Ri$ is the Richardson number, $Ri_c$ is a critical value for $Ri$ and $E_0$ is an amplitude parameter. $Ri$ is defined as $GH^3 \cos\theta / F_{sl}^2$ whereby $G = g\Delta\rho(z)/\rho_0$ is the buoyancy, $\Delta\rho(z)$ is the difference between downflow density and the mean local ambient density and $\theta$ is

the slope angle. Values for $Ri_c$ and $E_0$ are taken to be 0.25 and 0.20 respectively (Xu et al., 2006). Finally, $C_d$ and $S_2$ are the drag coefficient and a constant, respectively, with values taken from the literature cited above. Initial conditions at the shelf break taken to be $H_0 = 100\,\text{m}$ and $F_{sl,0} = H_0(G_0 H_0)^{0.5}$ with $G_0$ calculated as above using the Southern Ocean density at the shelf break depth. We integrate Eq. (25) and Eq. (26) every 2 meters from the shelf break until the depth where the plume and the ambient water buoyancy are close enough as defined by a threshold. Then this downslope flow enters Southern Ocean

model layers in accordance with the plume height. Although other overflow approaches have been proposed (Danabasoglu et al., 2010; Xu et al., 2006), we think that this simple and fast approach is well suited to capture the formation and insertion of AABW at depth and is a particular strength in our simplified model. In this context we note that, quite a few CMIP6 models form AABW incorrectly by deep, open-ocean convection and/or with lack of plume entrainment (Heuzé, 2021).

With the above physics and for any ocean zone, the general conservation equation of any ocean tracer $\psi$ may be written as

$$\frac{\partial\psi}{\partial t} + \frac{1}{a\cos\phi}\frac{\partial(\cos\phi\, v\psi)}{\partial\phi} + \frac{\partial(w\psi)}{\partial z} = \frac{1}{a^2\cos\phi}\frac{\partial}{\partial\phi}\left(\cos\phi\, K_h \frac{\partial\psi}{\partial\phi}\right) + \frac{\partial}{\partial z}\left(K_v \frac{\partial\psi}{\partial z}\right) + \Psi_S(\psi) + \Psi_B(\psi) + \Psi_I(\psi) \tag{27}$$

where $\Psi_S$ is the air-sea exchange of heat and gases, $\Psi_B$ is the exchange of dissolved substances with the ocean sediment and

$\Psi_I$ are internal source/sinks into the water column. Ocean tracers of temperature, salinity and $\delta^{18}O_w$ are forced only at the ocean surface via air-sea heat exchange and direct solar forcing for temperature and freshwater forcing for salinity and $\delta^{18}O_w$. Pacific and Atlantic Ocean model sectors between $35°$ S and $55°$ S south of Africa are connected via a zonal surface-intensified mixing scheme in order to emulate the role of Antarctic Circumpolar Current there. These specific terms are added to Eq. 27 for tracers of these boxes (msA and msP).

**2.4 Ocean biogeochemical cycling**

The ocean module considers the following biogeochemical ocean tracers: phosphate (PO$_4$), dissolved oxygen (O$_2$), dissolved inorganic carbon (DIC) in [12,13,14]C species, and alkalinity (ALK), which are all forced by new (export) production of organic matter and biogenic calcium carbonate shells in the lighted ocean surface layers. Furthermore, PO$_4$, DI[12,13,14]C and ALK are forced by river inputs and concentration/dilution of the surface layer by evaporation/precipitation. Moreover, O$_2$ and DI[12,13,14]C

are forced by air-sea exchange. In the ocean interior, all these tracers are influenced by remineralization of organic matter and dissolution of CaCO$_3$ shells in the water column as well as exchange with the ocean sediment. DI[14]C is affected by radioactive





decay in all ocean layers. For simplicity, we have neglected explicit nitrogen cycling and have assumed that all biogenic matter export from the surface layer is in the form of particle organic matter (POM) and that all $CaCO_3$ is in the form of calcite.

We take new (export) production of organic matter (NP) in each model surface layer to be a function of phosphorus (Maier-Reimer, 1993; Yamanaka and Tajika, 1996) and solar radiation as

$$\text{NP} = A_o^{if} \, z_{eu} \, L_f \, [\text{PO}_4] \frac{I}{I+I_{1/2}} \frac{[\text{PO}_4]}{[\text{PO}_4]+P_{1/2}} \tag{28}$$

where $A_o^{if}$ is the ice-free ocean surface area, $z_{eu}$ is the euphotic layer depth (100 m), $[\text{PO}_4]$ and $I$ are phosphate content in the surface ocean and solar radiation there, respectively. $P_{1/2}$ and $I_{1/2}$ are their respective half saturation constants equal to 1 $\mu$mol m$^{-3}$ for phosphate and 100 W m$^{-2}$ for light (Mutshinda et al., 2017). $L_f$ is an efficiency coefficient (units of s$^{-1}$) that estimates iron and/or other limitation factors on net primary production. We take the value of $L_f$ to be equal to 1 for most ocean zones but set to some lower value for Southern and Arctic Oceans as determined by model fit to ocean data (Sect. 3.1.2). In the surface layer sources/sinks due to new production for $\text{PO}_4$, $\text{DI}^{12}\text{C}$, ALK and $O_2$ are $-\text{NP}$, $-r_{\text{CP}}\text{NP}$, $r_{\text{AlkP}}\text{NP}$ and $(r_{\text{OCP}} + r_{\text{ONP}})\text{NP}$ respectively, where $r_{\text{CP}}$, $r_{\text{AlkP}}$, $r_{\text{OCP}}$ and $r_{\text{ONP}}$ are the Redfield ratios of C:P, ALK:P, $(O_2)_C$:P and $(O_2)_N$:P respectively. The subscripts C and N refer to a division of POM produced into "carbon" and "nutrient" parts respectively, as explained below. We adopted the Redfield ratios used in Shaffer et al., (2008) and shown in Table 2. For $\text{DI}^{13,14}\text{C}$, the surface sink due to new production and associated isotope fractionation is $-^{13,14}\alpha_{\text{Org}}\left([\text{DI}^{13,14}\text{C}]/[\text{DI}^{12}\text{C}]\right)_{eu} r_{\text{CP}}\text{NP}$. We take the fractionation factor $^{13}\alpha_{\text{Org}}$ to depend on surface ocean concentrations of dissolved carbon dioxide ($[\text{CO}_{2(aq)}]$) and phosphate according to Pagani et al., (1999):

$$^{13}\alpha_{\text{Org}} = 25 - \frac{116.96 \, L_f \, [\text{PO}_4] + 81.42}{[\text{CO}_{2(aq)}]} \tag{29}$$

where concentrations are in $\mu$mol kg$^{-1}$. As previously, $^{14}\alpha_{\text{Org}} = 1 - 2\left(1 - \,^{13}\alpha_{\text{Org}}\right)$.

Surface (export) production of biogenic calcite is related to new (export) production by $r_{\text{CalC}} \, r_{\text{CP}} \, \text{NP}$ where $r_{\text{CalC}}$ is the "rain" ratio, the ratio between the production of $CaCO_3$ to the production of organic carbon. It is parameterized according to Maier-Reimer (1993) and Marchal et al., (1998) but with the addition of a dependence on the calcite saturation state of the ocean surface layer ($\Omega_S$) following Shaffer et al., (2016):

$$r_{\text{CalC}} = r_{\text{Cal,max}} \frac{\exp\left(\mu(T_s - T_{ref})\right)}{1 + \exp\left(\mu(T_s - T_{ref})\right)} \frac{\Omega_S - 1}{\nu + (\Omega_S - 1)} \tag{30}$$





where $r_{Cal,max}$ is a rain ratio upper limit, $\mu$ is the steepness factor, $T_s$ and $T_{ref}$ are the surface temperature and the reference temperature (10 °C) respectively, and ν is a half saturation constant taken to be 1 (Gangstø et al., 2011). Furthermore, $\Omega_S = [Ca^{2+}][CO_3^{2-}]/K_{sp}$ with calcium concentration given as $[Ca^{2+}] = [Ca^{2+}]_m(S/S_m)$ where $[Ca^{2+}]_m$ and $S_m$ are the global ocean mean calcium and salinity values, taken as 10.57 mol m⁻³ and 35 for present day respectively, S is the ocean salinity and $K_{sp}$ is the calcite solubility coefficient. There is no biogenic calcite production for subsaturated conditions ($r_{calC} = 0$ for $\Omega_S \leq 1$). Values for $r_{Cal,max}$ and $\mu$ are determined by model fit to ocean and ocean sediment data. With this, surface sinks for DI¹²C and ALK due to biogenic calcite production are $-r_{calC}\,r_{CP}\,\mathrm{NP}$ and $-2r_{calC}\,r_{CP}\,\mathrm{NP}$ respectively. For DI¹³,¹⁴C, the surface sinks due to calcite production and associated isotope fractionation are $-^{13,14}\alpha_{cal}([DI^{13,14}C]/[DI^{12}C])_{eu}r_{calC}\,r_{CP}\,\mathrm{NP}$, where $\alpha_{cal} = 1$, assuming no carbon fractionation during biogenic calcite formation in the ocean surface layer.

Particles are assumed to sink out of the surface layer with settling speeds high enough to neglect advection and diffusion of them. This particulate flux decreases significantly with depth due to subsurface remineralization/dissolution with only a small fraction reaching the sea floor as shown by sediment trap data (Martin et al., 1987). To address this, we assume an exponential-type law for the vertical fraction of particulate organic matter (POM) "nutrient" and "carbon" component, each with a distinct e-folding length ($\xi_N$ and $\xi_C$) motivated mainly by results of Shaffer et al., (1999). Additionally, we also include temperature dependence ($\lambda_Q$) on remineralization rates as indicated by ocean data (Laufkotter et al., 2017; Marsay et al., 2015). Therefore, tracer sources in the water column due to remineralization of POM, "nutrient" ($\Phi_N$) and "carbon" component ($\Phi_C$), are expressed as the vertical gradient of POM,

$$\Phi_{N,C}(z) = \frac{\partial \mathrm{POM}(z)_{N,C}}{\partial z} = \mathrm{POM}(z)_{N,C}\,\frac{\lambda_Q(T_o(z))}{\xi_{N,C}} \tag{31}$$

where $\lambda_Q$ is defined as $Q_{10}^{(T_o(z)-T_{o,ref})/10}$, whereby $Q_{10}$ is the biotic activity increase for a 10 degree increase of $T_o$, $T_{o,ref}$ is a reference temperature taken as the present-day global area-weighted mean observed temperature from the World Ocean Atlas 2018 database (Boyer et al., 2018) for the upper 500 m (Komar and Zeebe, 2021) and $\xi_{N,C}$ are e-folding lengths for $T_o(z) = T_{o,ref} = 9.3$ °C. For the biogenic calcium carbonate particles flux (PCal), we take a simple exponential law with a constant e-folding length $\xi_{Cal}$ and, as above, tracer sources in the water column due to dissolution of CaCO₃ produced in the euphotic zone ($\Phi_{Cal}$) is expressed as

$$\Phi_{Cal}(z) = \frac{\partial \mathrm{PCal}(z)}{\partial z} = \frac{\mathrm{PCal}(z)}{\xi_{Cal}} \tag{32}$$





In Eqs. 31 and 32, $\text{POM}_N$, $\text{POM}_C$ and PCal at $z = 0$ are the respective surface layer export productions, NP, $r_{CP}$ NP
and $r_{calc} r_{CP}$ NP. Given the range of $Q_{10}$ values shown by data and modelling studies (Laufkotter et al., 2017; Regaudie-de-
Gioux and Duarte, 2012; Bendtsen et al., 2015) and to maintain model simplicity (see Sect. 2.5 and Sect. 2.6) we choose a
constant value for $Q_{10}$ of 2. Also for simplicity, $\xi_N$, $\xi_C$ and $\xi_{Cal}$ are taken to be constants for all ocean zones whose values
(Table 2) has been chosen to fit ocean data (Sect. 3.1.2). Thus, the vertical source/sinks in the ocean interior for $\text{PO}_4$, $\text{DI}^{12}\text{C}$,
ALK and $\text{O}_2$ are $\Phi_N$, $(\Phi_C + \Phi_{Cal})$, $(2\,\Phi_{Cal} - r_{AlkP}\,\Phi_N)$ and $(r_{ONP}\,\Phi_N + r_{OCP}\,\Phi_C)$ respectively. For $\text{DI}^{13,14}\text{C}$, the vertical
distribution from remineralization and dissolution is $\left([\text{DI}^{13,14}\text{C}]/[\text{DI}^{12}\text{C}]\right)_{eu}\left[\,^{13,14}\alpha_{Org}\Phi_C + ^{13,14}\alpha_{Cal}\Phi_{Cal}\right]$. All these vertical
distributions are weighted by the ocean area profile $A_o(z)$ for each zone. In addition, the fluxes of P and $^{12,13,14}\text{C}$ that fall in
the form of POM and/or biogenic calcite particles on the model ocean sediment surface at any depth of each zone are calculated
as the product of $dA_o(z)/dz$ there and the difference between the particulate fluxes falling out of the ocean surface layer and
the remineralization/dissolution taking place down to the depth of each zone.


Non-linear ocean carbonate chemistry is calculated using the recursive formulation of Antoine and Morel (1995) as explained
in detail in Shaffer et al., (2008) in the context of a DCESS-model approach. This system yields ocean distributions of $\text{CO}_{2(aq)}$,
$\text{CO}_3^{2-}$, $\text{HCO}_3^-$ and hydrogen ion concentrations needed for calculations of air-sea exchange of carbon dioxide, carbon isotopic
fractionation during air-sea exchange and in ocean new production, dissolution of calcite in the ocean sediment, and pH
(seawater scale) calculations. Profiles of carbonate saturation with respect to calcite are calculated as $\text{K'}_{CaCO3}/([\text{Ca}^{2+}]_m S/S_m)$,
where $\text{K'}_{CaCO3}$ is the apparent dissociation constant for calcite as function of T, S and pressure (Mucci, 1983).

## 2.5 Ocean Sediment

For the sediment module, we adopt the approach developed by Shaffer et al., (2008), the main features of which are
summarized below.


Each model ocean layer of 100 m thickness is assigned a sediment segment composed of calcite, non-calcite mineral (NCM)
and reactive organic matter. The segment is a bioturbated layer (BL) that is assumed to be 10 cm thick divided in 7 sublayers
with highest resolution near the sediment surface such that sublayer boundaries are 0, 0.2, 0.5, 1, 1.8, 3.2, 6 and 10 cm.
Sediment segment areas are determined by model topography from each ocean zone. POM and PCal rain fluxes, and ocean
values of T, S, DIC, ALK, $\text{O}_2$ and $\text{PO}_4$ are taken from respective layers from ocean and ocean biogeochemistry modules. NCM
fluxes ($F_{NCM}$) are parameterized as

$$F_{NCM} = \text{NCF}\left[1 + \text{CAF}\exp\left(-z/\lambda_{slope}\right)\right] \tag{33}$$





where NCF is the open ocean non-calcite flux, CAF is an amplification factor at the coast (i.e. at $z = 0$) and $\lambda_{slope}$ is the e-folding, water depth scale representing the effect of distance from the coast associated with continental slope topography. For simplicity, we apply the same value of NCF, CAF and $\lambda_{slope}$ for all ocean sectors taken to be 0.3 g cm$^{-2}$ kyr$^{-1}$, 20 and 200 m respectively.

The sediment module is designed to address calcium carbonate (CaCO$_3$) dissolution and (oxic and anoxic) organic matter remineralization by calculating concentrations of reactive organic carbon (OrgC), pore-water O$_2$ and pore-water CO$_3^{2-}$ for each sediment sublayer. To accomplish this, a key property is the sediment porosity ($\phi_S$, not to be confused with the latitude symbol), which is parameterized as a function of calcite dry weight fraction, (CaCO$_3$)$_{dwf}$:

$$\phi_S(\zeta) = \phi_{S,min} + (1 - \phi_{S,min})\exp(-\zeta/\alpha) \tag{34}$$

where $\zeta$ is the sediment vertical coordinate, $\phi_{(S,min)} = 1 - (0.483 + 0.45(CaCO_3)_{dwf})$ and $\alpha = 0.25(CaCO_3)_{dwf} + 3(1 - (CaCO_3)_{dwf})$ following Archer (1996). From sediment porosity, the sediment formation factor ($F_S = \phi_S^{-3}$) is determined in order to calculate bulk sediment diffusion coefficients of pore-water solutes.


To account for the role of benthic fauna, the bioturbation rate ($D_b$) is parameterized to depend on organic carbon rain rates but also to consider attenuation associated with very low dissolved oxygen concentrations. This is formulated as

$$D_b = D_b^0 \left(\frac{F_{OrgC}}{F_{OrgC}^0}\right)^{0.85} \frac{[O_{2,ocean}]}{[O_{2,ocean}] + O_{2,low}} \tag{35}$$


where $[O_{2,ocean}]$ is the ocean O$_2$ concentration at the sediment surface and O$_{2,low}$ is taken to be 20 mmol m$^{-3}$. Moreover, $D_b^0$ and $F_{OrgC}^0$ are the bioturbation rate scale and the organic carbon rain rate scale whose values are $1.38 \times 10^{-8}$ cm$^2$ s$^{-1}$ and $1 \times 10^{-12}$ mol cm$^{-2}$ s$^{-1}$, based in part on Archer et al., (2002).

Oxygen remineralization rates in the BL are taken to scale as bioturbation rates (and thereby as organic carbon rain rates; Archer et al., (2002)), such as $\lambda_{ox} = \lambda_{ox}^0 D_b/D_b^0$. Anoxic remineralization rates in the BL are slower than oxic rates and will depend upon the specific remineralization reactions involved (e.g. denitrification faster than sulphate reduction). More organic rain would be associated with a more anoxic BL and a shift toward sulphate reduction. Therefore, we take $\lambda_{anox} = \beta\lambda_{ox}$ whereby $\beta$ is taken to decrease for increasing organic carbon rain rate such that $\beta = \beta_0 (F_{OrgC}/F_{OrgC}^0)^\gamma$. As described in 555    Shaffer et al., (2008), values for $\lambda_{ox}^0$, $\beta_0$ and $\gamma$ were constrained by organic carbon burial observations to be $1 \times 10^{-9}$ s$^{-1}$, 0.1 and -0.3 respectively.





Governing equations for OrgC, pore-water $O_2$ and $CO_3^{2-}$ and $CaCO_3$ are second-order, non-linear coupled differential equations which are solved for each sediment segment using a semi-analytical iterative approach (steady state) or time stepping approach (time dependent) by imposing boundary conditions at the top and bottom of the BL and matching conditions at the sublayer boundaries. For simplicity we also apply the same calculated sediment remineralization rates to organic phosphorus raining on the sediment surface. Using a mass balance approach, sedimentation velocity is determined and used to calculate burial rates of phosphorus, organic carbon and carbonate carbon down out of the base of the BL. In this way the model produces synthetic sediment cores at every depth for each of the model ocean basins. Furthermore, model solutions provide fluxes between ocean and sediment layers of $PO_4$, $O_2$, DIC and ALK based on concentrations of these tracers in each respective adjacent ocean layer and sediment pore-water concentrations from organic matter remineralization and calcium carbonate dissolution. A detailed description of the sediment module is given in Appendix A of Shaffer et al., (2008).

**2.6 Land Biosphere**

We consider a land biosphere model with three different, dynamically varying vegetation zones per hemisphere (tropical forest, TF, grasslands, savanna and deserts, GSD, and extratropical forest, ET) that include carbon reservoirs for leaves ($M_G$), wood ($M_W$), litter ($M_D$) and soil ($M_S$) (Shaffer et al., 2008; Eichinger et al., 2017). The meridional limits for each vegetation zone ($\phi_{TF-GSD}$ for TF-GSD boundary and $\phi_{GSD-EF}$ for GSD-EF boundary) are obtained using a fifth-order polynomial dependent on the deviation of hemispheric annual mean atmospheric temperature from the respective pre-industrial temperature such that $\phi(\delta T_a) = c_1 \delta T_a^5 + c_2 \delta T_a^4 + c_3 \delta T_a^3 + c_4 \delta T_a^2 + c_5 \delta T_a^1 + c_6$, with $\delta T_a = \overline{T}_a - \overline{T}_{a,PI}$. Polynomial coefficients are obtained by fitting data from Gerber et al., (2004) for Northern and Southern Hemisphere separately (Table 3). Poleward ET limits are the annual mean snowline for Northern Hemisphere and the fixed position at 55° S for Southern Hemisphere (there is no land south of 55° S, see Fig. 1a).





**Table 3: Coefficients for vegetation meridional limits, $\phi(\delta T_a)$, for Northern and Southern Hemispheres and global pre-industrial distribution of carbon storage and net primary production for all vegetation zones considered in the model.**

|  |  | $c_1$ ($\times 10^{-5}$) | $c_2$ ($\times 10^{-4}$) | $c_3$ ($\times 10^{-3}$) | $c_4$ ($\times 10^{-2}$) | $c_5$ ($\times 10^{-1}$) | $c_6$ |
|---|---|---|---|---|---|---|---|
| $\phi_{TF-GSD}$ | NH | -1.803 | -5.809 | -5.168 | 4.970 | 10.920 | 11.280 |
|  | SH | 8.413 | 7.339 | -8.333 | -8.764 | -6.965 | -19.660 |
| $\phi_{GSD-EF}$ | NH | 1.152 | -1.785 | -4.557 | 4.156 | 10.170 | 37.770 |
|  | SH | -0.651 | 2.131 | 1.857 | -4.100 | -6.615 | -35.630 |

|  | Tropical forest | Grassland, Savanna and Deserts | Extratropical forests |
|---|---|---|---|
| Leaves (Gt C) | 30 | 50 | 20 |
| Wood (Gt C) | 270 | 50 | 180 |
| Litter (Gt C) | 16 | 40 | 64 |
| Soil (Gt C) | 200 | 500 | 800 |
| NPP (Gt C yr$^{-1}$) | 25 | 20 | 15 |

Net primary production on land (NPP) takes up atmospheric CO₂ and is forced by seasonally varying solar radiation. For each vegetation type zone, land NPP is calculated according to

$$\text{NPP} = \text{NPP}_{\text{PI}}\, A_f\, f(I) \left[ 1 + f_{CO_2} \ln\left( \frac{p\text{CO}_2}{p\text{CO}_{2,\text{PI}}} \right) \right] \tag{36}$$

where $\text{NPP}_{\text{PI}}$ is the pre-industrial net primary production (see Table 3), $A_f$ is an area factor accounting for vegetation size change with respect to pre-industrial size, $f_{CO_2}$ is the CO₂ fertilization factor equal to 0.37, a suitable value for the terrestrial biosphere (Eby et al., 2013; Zickfeld et al., 2013), $p\text{CO}_2$ is the model-calculated partial pressure of atmospheric carbon dioxide, $f(I) = f_0 + a \exp\left( -\left((I-b)/c\right)^2 \right)$ is a function fitted from model results (Hazarika et al., 2005) with coefficients $a$, $b$ and $c$ chosen to represent the seasonal cycle of solar radiation according to the specific orbital forcing parameters and $f_0$ is chosen so that the annual mean value of $f(I)$ for each zone equal to one. With this formulation NPP responds in a seasonal cycle according to $f(I)$ and to atmospheric carbon dioxide on longer time scales.

With the descriptions above, the conservation equations for the land biosphere reservoirs of ¹²C for leaves ($M_G$), wood ($M_W$), litter ($M_D$) and soil ($M_S$) for each of the six vegetation zones are:

$$\frac{dM_G}{dt} = \frac{35}{60}\,\text{NPP} - \frac{35}{60}\,\text{NPP}_{\text{PI}}\,\frac{M_G}{M_{G,\text{PI}}} \tag{37}$$

$$\frac{dM_W}{dt} = \frac{25}{60}\,\text{NPP} - \frac{25}{60}\,\text{NPP}_{\text{PI}}\,\frac{M_W}{M_{W,\text{PI}}} \tag{38}$$





$$\frac{dM_D}{dt} = \frac{35}{60} \text{NPP}_{\text{PI}} \frac{M_G}{M_{G,\text{PI}}} + \frac{20}{60} \text{NPP}_{\text{PI}} \frac{M_W}{M_{W,\text{PI}}} - \frac{55}{60} \text{NPP}_{\text{PI}} \frac{M_D}{M_{D,\text{PI}}} \lambda_Q \qquad (39)$$

$$\frac{dM_S}{dt} = \frac{5}{60} \text{NPP}_{\text{PI}} \frac{M_W}{M_{W,\text{PI}}} + \frac{10}{60} \text{NPP}_{\text{PI}} \frac{M_D}{M_{D,\text{PI}}} \lambda_Q - \frac{15}{60} \text{NPP}_{\text{PI}} \frac{M_S}{M_{S,\text{PI}}} \lambda_Q \qquad (40)$$

where $M_{G/W/D/S,PI}$ are the pre-industrial reservoir sizes (Table 3) and $\lambda_Q = Q_{10}^{(T_a - T_{a,PI})/10}$ with $Q_{10} = 2$ as above (Sect. 2.4). Atmosphere-land biosphere carbon dioxide flux from each vegetation zone is:


$$F_{CO_2} = -\text{NPP} + \text{NPP}_{\text{PI}} \left[ \frac{45}{60} \frac{M_D}{M_{D,\text{PI}}} + \frac{15}{60} \frac{M_S}{M_{S,\text{PI}}} \right] \lambda_Q \qquad (41)$$

For isotopes $^{13}$C and $^{14}$C, Eqs. (37)-(41) are extended considering fractionation factors for photosynthesis and, for $^{14}$C, radioactive decay as described in Shaffer et al., (2008).


Finally, land biosphere methane and nitrous oxide productions ($F_{CH_4}$ and $F_{N_2O}$ respectively) take place in soil and are proportional to the reservoir size and temperature dependent according to $\lambda_Q$ where again $Q_{10} = 2$ for both. In addition, we assume methane emissions only from wet areas (zones TF and ET). Thus, for each vegetation zone fluxes of these two greenhouse gases are given as


$$F_{CH_4|N_2O} = F_{CH_4|N_2O,PI} \frac{M_S}{M_{S,PI}} \lambda_Q \qquad (42)$$

where $F_{CH_4|N_2O,PI} = \tau_{CH_4|N_2O,PI} (pCH_4|pN_2O)_{PI}$ and $\tau_{CH_4|N_2O,PI}$ is the pre-industrial atmospheric life time of CH$_4$ and N$_2$O (9.5 and 150 years, respectively). $^{13,14}$CH$_4$ fluxes are $^{13,14}\alpha_M F_{CH_4,PI} (M_S^{13,14}/M_{S,PI}) \lambda_Q$ with $^{13}\alpha_M = 0.97$ the fractionation factor

for CH$_4$ production. As above, we assume the $^{14}$C/$^{12}$C fractionation to be twice as strong as for $^{13}$C/$^{12}$C (see Sect. 2.2). Fluxes to specific atmospheric boxes are equal to the biosphere fluxes within the latitudinal boundaries of the boxes.

### 2.7 Lithosphere module (rock weathering, volcanism, and river input)

We follow the same approach as in the DCESS I model (Sect. 2.7 in Shaffer et al., (2008)) by considering river inputs of phosphorus and carbon species, climate-dependent carbonate and silicate weathering rates and lithosphere outgassing.

However here we extend this approach to consider distributions of continents, river mouths (Dai and Trenberth, 2002) and volcanoes (NCEI Volcano Location Database). In the following, we restrict ourselves to presenting the main features of this module (see Shaffer et al., (2008) again for more details).





Weathering rates of rocks containing phosphorus ($W_P$), as well as carbonate and silicate weathering rates ($W_{Sil}$ and $W_{Cal}$) are taken to depend on deviation of mean atmospheric temperature from its pre-industrial value in the form,

$$W_{P|Sil|Cal} = \lambda_Q W_{P|Sil|Cal,PI} = Q_{10}^{(T_a - T_{a,PI})/10} W_{P|Sil|Cal,PI} \tag{43}$$

where $W_{P|Sil|Cal,PI}$ are the pre-industrial weathering rates for phosphorus, silicate and carbonate and $Q_{10} = 2$ as for the other model components. Silicate weathering consumes 2 moles of atmospheric $CO_2$ per mole of silicate mineral weathered while the carbonate weathering consumes only 1 mole of atmospheric $CO_2$ per mole of carbonate mineral weathered. Both types of weathering supply bicarbonate ion to ocean surface layers, modifying dissolved inorganic carbon and alkalinity concentrations. The phosphorus supply is equal to $W_P$. Therefore, expressions for total river inputs for PO4, DIC and ALK tracers are

$$R_P = W_P = \lambda_Q W_{P,PI} \tag{44}$$

$$R_{DIC} = 2(W_{Sil} + W_{Cal}) = 2\lambda_Q(W_{Sil,PI} + W_{Cal,PI}) \tag{45}$$

$$R_{ALK} = 2(W_{Sil} + W_{Cal}) - r_{ALKP} W_P = \lambda_Q[2(W_{Sil,PI} + W_{Cal,PI}) - r_{ALKP} W_{P,PI}] \tag{46}$$

Values of $W_{P,PI}$, $W_{Cal,PI}$ and $W_{Sil,PI}$ are obtained from the assumed pre-industrial steady state equal to the global ocean burial rate of phosphate ($B_{OrgP}$) and carbonate ($B_{Cal}$), respectively, and the assumption that $W_{Sil,PI}$ can be taken to be a fixed ratio of carbonate weathering: $W_{Sil,PI} = \gamma_{Sil} W_{Cal,PI}$ with $\gamma_{Sil} = 0.85$. These total river inputs are distributed among the twelve ocean sectors according to river mouth distributions mentioned above. For example, this leads to no river input to the model Southern Ocean sector.

Sources of carbon to the atmosphere are weathering of rocks containing old organic carbon ($W_{OrgC}$) and lithosphere outgassing ($Vol$). As above, $W_{OrgC} = \lambda_Q W_{OrgC,PI}$ and $Vol$ may either be taken as a constant and equal to its pre-industrial value ($Vol_{PI}$) or may be prescribed as an external forcing of the Earth System. Given the above together with assigned or calculated $^{13}$C content for the different model inputs and outputs (including for example for organic carbon burial), overall steady state conservation equations can be formulated for both $^{12}$C and $^{13}$C. These conservation equations can then be used to derive expressions for $W_{OrgC,PI}$ and $Vol_{PI}$ as given in Shaffer et al., (2008) (their Eqs. (41) and (42)). These total atmospheric carbon inputs are distributed among the 6 atmosphere sectors according to the same meridional distribution as carbonate and silicate weathering and volcano distribution mentioned above.

From the above, the pre-industrial steady state equations for phosphorus, carbon-12 and carbon-13 are given by





$$W_P - B_{OrgP} = 0 \tag{47}$$

$$W_{OrgC} + Vol - \frac{\gamma_{Sil}}{1+\gamma_{Sil}} B_{Cal} - B_{OrgC} = 0 \tag{48}$$

$$W_{OrgC}\delta^{13}C_{OrgC} + Vol\,\delta^{13}C_{Vol} - \frac{\gamma_{Sil}}{1+\gamma_{Sil}} B_{Cal}\delta^{13}C_{Cal} - B_{OrgC}\delta^{13}C_{OrgC} = 0 \tag{49}$$

For total oxygen content in the ocean-atmosphere system, the pre-industrial steady state equation is

$$\frac{r_{OCP}}{r_{CP}}\big[B_{OrgC} - W_{OrgC} - f_{old,OM}\,Vol\big] + r_{ONP}\,(B_P - W_P) = 0 \tag{50}$$

where $r_{ONP}$, $r_{OCP}$ and $r_{CP}$ are as mentioned in Sect. 2.4, and $f_{old,OM}$ is the fraction of $Vol$ originating from old organic matter

that results from the above conservation calculations for $^{12}$C and $^{13}$C.

## 3 Model solution, calibration and validation

### 3.1 Pre-industrial steady state solution

#### 3.1.1 Solution procedure

The ocean module equations are discretized on a staggered grid type, with tracer values defined at the centre of boxes and

velocities and diffusivities determined at box edges. Centered differences are used for derivatives, diffusion and vertical advection, whereas an upwind scheme is used for the coarsely resolved, meridional advection. Prognostic equations for the atmosphere (including snow and sea ice cover), land biosphere, lithosphere and the ocean modules are solved simultaneously using a fourth order Runge-Kutta algorithm with a two week time step. Prognostic equations for the ocean sediment are solved by simple time stepping with a one year time step. The complete, coupled model is written in Fortran language and runs at a

speed of about 10 kyr of simulation per 30 minutes of computer time on a high-end personal computer.

#### 3.1.2 Calibration procedure

For model calibration we used a similar approach as in Shaffer et al., (2008) but here consisting of six steps. For the first step we considered the atmospheric module with atmospheric $p$CO$_2$, $p$CH$_4$ and $p$N$_2$O set to their pre-industrial values (280, 0.72 and 0.27 $\mu$atm respectively) and a slab ocean for air-sea heat exchange. We adjusted the free parameters listed in Table 1 to

give a steady state global annual mean atmospheric temperature of 15 °C and a climate sensitivity of 3 °C per doubling CO$_2$, a poleward transport of heat and water vapor consistent with observations and annual mean latitudes and seasonal cycle amplitudes of sea ice lines in accordance with observed data. In the second step we couple the physics part of the ocean module to the physics part of atmosphere module and physical ocean free parameters (Table 2) were adjusted in order to get the best fit to observed mass and heat transport as well as temperature and salinity distributions. In the third step we couple the land



biosphere to the previously calibrated "physics model" version, and we adjust free parameters of function $f(I)$ in Eq. (36) to give observed annual mean carbon reservoirs for each vegetation zones as well as an atmospheric carbon dioxide seasonal cycle amplitude consistent with observations. In the fourth step we incorporate ocean biogeochemical tracers PO$_4$, DI$^{12,13,14}$C, ALK and O$_2$ to the model version of step three. For calibration we start with homogeneous vertical values of $2.17 \times 10^{-3}$, 2.32 and 2.43 mol m$^{-3}$ for PO$_4$, DIC and ALK respectively (in the following for simplicity DIC will be used to mean DI$^{12}$C).

Furthermore, in this step we use a fixed atmospheric O$_2$ equal to 0.2095 $\mu$atm, an atmospheric $\delta^{13}$C equal to -6.5 ‰ and an atmospheric $^{14}$C production chosen to keep $\Delta^{14}$C$_{atm} \sim$ 0 ‰ (Sarmiento and Gruber, 2006; Shaffer et al., 2008). At this stage we assume that all biogenic particles falling to the ocean bottom remineralize completely there. We then made initial guesses for the values of the biogeochemical free parameters of the ocean module listed in Table 2. These choices were partially based on DCESS I model (Shaffer et al., 2008). The atmosphere-land biosphere-ocean model was spun up with uniform atmosphere

and ocean tracer distributions to a steady state after about 10,000 model years and results were compared with atmosphere, land biosphere and ocean data. Then all parameter values were adjusted by trial and error in order to get steady state solutions that better satisfied requirements described in the previous steps as well as observed global annual mean distributions of T, S, PO$_4$, DIC, ALK and dissolved O$_2$.

In the fifth calibration step we coupled the sediment module to the step four calibrated model. For conservation, the total burial rate of a tracer was added to the ocean surface layer of each box under consideration of present-day rivers distribution. After solving this new closed-system for a new, pre-industrial steady state, we adjusted all free parameters in order to obtain steady state solutions that better satisfied the data-based constrains of the previous calibration steps. In the sixth and final calibration step, we coupled the lithosphere module to the step five calibrated model whereby river inputs are equated with tracer burial

fluxes from the fifth calibration step (tracer burial fluxes are now leaving the system). In addition, weathering rates and lithosphere outgassing were calculated from the tracer burial fluxes and the assumption of the pre-industrial steady state for phosphorus and $^{12,13}$C (Sect. 2.7). A last slight trial and error adjustment is made to satisfy the data-based requirements from all previous calibration steps. For this final calibration, we made a long run until steady state is achieved such that all model components vary less than a 0.001 % during 1000 model years. Resulting global annual means for atmospheric temperature

and CO$_2$, CH$_4$ and N$_2$O atmospheric partial pressures are 15.12 °C, 279.96, 0.72 and 0.27 $\mu$atm respectively. For atmospheric isotopes, this final calibration gives a global mean atmospheric $\delta^{13}$C of -6.53 ‰ and a $^{14}$C atmospheric production of $1.66 \times 10^4$ atom m$^{-2}$ s$^{-1}$. Global mean ocean values are 2.16 and 145.07 mmol m$^{-3}$ for PO$_4$ and dissolved O$_2$ and 2.30 and 2.41 mol m$^{-3}$ for DIC and ALK respectively.

### 3.1.3 Atmosphere tracers and transport results

The steady-state, pre-industrial solution gives the annual mean atmospheric temperatures of 15.4 °C for Northern Hemisphere and 14.8 °C Southern Hemisphere, whereas annual mean sea ice extensions for Northern (Arctic and North Pacific sectors) and Southern Hemispheres are 66.2° N, 64.9° N and 63.2° S respectively. While Southern Ocean model result is close to the





observed value (~64° S), our Arctic Ocean sea ice line position is about 7° too far south with respect to observational estimates (Fetterer et al., 2017). This could be attributable to ocean and sea ice dynamics that are not captured in our simplified, zonal-averaged model. The Northern Hemisphere snow line is found at 58.8° N. Annual mean poleward atmospheric heat transport across 35° N/S and 55° N/S are 4.4/4.8 and 3.1/3.4 PW, and poleward water vapor transports in the atmosphere at the same latitudes are 0.76/0.82 and 0.43/0.45 Sv respectively. These model results agree well observational estimates (Trenberth and Caron, 2001). The seasonal cycle of sea ice is relatively well represented into the model, with a maximum (minimum) extension at the end of winter (summer) and with amplitudes of 4.7 and 9.0 degrees for Arctic and Southern Ocean respectively with differences of 0.4 and 0.9 degrees respective to observed estimates (Fetterer et al., 2017). We believe that model-data disagreements in sea ice are due to the simplicity of parameterization, but also due to anthropogenic signal in the modern-day sea-ice data. Annual mean model difference in atmospheric $pCO_2$ between Northern and Southern Hemispheres is only 0.7 $\mu$atm, a somewhat lower value than observations. However, much of this difference could be explained by the effect on the observations of NH anthropogenic $CO_2$ emissions. On the other hand, the atmospheric $pCO_2$ seasonal cycle amplitude of 4.6 and 0.9 ppm for Northern and Southern Hemisphere respectively agrees rather well with monthly-mean observations from Mauna Loa and Cape Grim observatories. This annual cycle amplitude responds strongly to the annual land vegetation dynamics as well as the ocean-land distribution between Northern and Southern Hemispheres. The Southern Ocean plays a role here as well, by dampening Southern Hemisphere cycle amplitude.

### 3.1.4 Ocean circulation and heat transport results

The steady state, pre-industrial model has the large-scale ocean circulation shown in Fig. 3. There are two meridional cells in the model Atlantic Ocean. The upper, clockwise cell is an Atlantic Meridional Overturning Circulation (AMOC) with a maximum transport of 19 Sv near 1000 m depth in the northern North Atlantic and a maximum depth of about 3500 m. Below 2000 m depth the flow returns southward to the Southern Ocean where upwells below the Drake Passage sill depth (2000 m), largely in response to the surface northward Ekman transport at 55° S. Both the intensity and depth penetration of modelled AMOC are in line with observed data and complex model results (Talley et al., 2003; Hirschi et al., 2020). The lower, counterclockwise cell carries mainly AABW, which fills the whole abyssal Atlantic. The upper, southward branch of this cell returns to the Southern Ocean and upwells there.



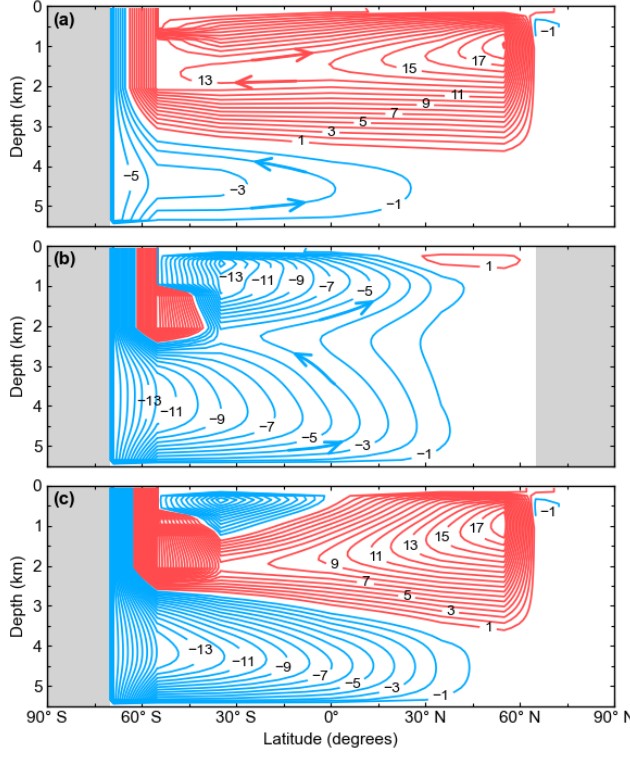

**Figure 3.** Pre-industrial, steady state model representation of the meridional overturning circulation (Sv) for: **(a)** the Atlantic Ocean, **(b)** the Pacific Ocean and **(c)** the global ocean. Red contours (positive values) represent clockwise circulation and blue contours (negative values) are for counterclockwise circulation as indicated schematically by arrows.

The Pacific Ocean overturning circulation is dominated by a counterclockwise cell carrying AABW which fills the whole deep ocean. The lower branch of the northward flow in this cell upwells in the North Pacific and returns southward in a near surface flow. The upper branch of the northward flow in this cell upwells and return southward to the Southern Ocean below 2000 m without crossing the equator. There is also a second counterclockwise cell carrying Antarctic Intermediate Water (AAIW) north of the equator above 2000 m depth. The upper, southward branch of this cell joins the southward, near surface flow of the deep, counterclockwise cell described above. There is a weak clockwise cell confined above 500 m depth in the North Pacific related in part to the imposed model outflow to the Arctic Ocean through Bering Strait.

The mediterranean Arctic Ocean presents an estuarine-type meridional circulation of 1 Sv intensity entering at 700 m depth and flowing out at 300 m depth. This counterclockwise circulation is maintained mainly by the zonal-averaged geometry and freshwater inputs from precipitation and runoff. The Southern Ocean shelf produces 5.4 Sv of overflow water which after entrainment increases its volume to ~16 Sv upon outflow in the deep Southern Ocean. This agrees quite well with observed estimates (Orsi, 2002; Gordon, 2019). Such realistic entrainment and subsequent AABW formation, as achieved in the model





with the prescribed, present day Antarctic continental slope (Amante and Eakins, 2009), underlines the usefulness of our simplified, gravity current approach to this problem.

Northward ocean heat transport is found throughout the Atlantic Ocean, peaking at 35° N with 0.83 PW and falling subsequently to 0.75 and 0.19 PW at 55° N and 65° N respectively. The high value at 55° N is related to the maximum ocean circulation intensity there. The South Atlantic carries 0.37 and 0.34 PW northward at 35° S and 55° S respectively. The North Pacific Ocean transports heat northwards, with values of 0.41 and 0.29 PW at 35° N and 55° N respectively. The South Pacific Ocean transports heat southwards, peaking at 35° S with 1.21 PW and 0.84 PW at 55° S. At the equator, the Atlantic and the

Pacific Oceans transport heat in opposite directions with 0.65 PW northwards and 0.37 PW southwards respectively. All modelled ocean transports agree well with data-based estimates (Trenberth and Caron, 2001), except at 55° N where the modelled value is greater than observations.

### 3.1.5 Ocean tracer and biological production results

Model ocean profiles of T, S and $\delta^{18}O_w$, are plotted together with observational data in Fig. 4 (all observed data shown in Figs.

4-7 are from the Global Ocean Data Analysis Project version 2.2022 (GLODAPv2, Lauvset et al., (2022)), except for those of $\delta^{18}O_w$ where data from the Global Seawater Oxygen-18 Database – v1.22 (Schmidt, 1999; Bigg and Rohling, 2000) have been used. In general, there is a good model-data agreement especially at mid-depth and in the deep and abyssal ocean. In the upper ocean, the model profiles are well within the observed mean range values. In particular for temperature, the model captures quite well surface-to-deep ocean transitions in most of the model ocean sectors. Modelled temperatures in the Arctic Ocean

below 1000 m are about 3.5 °C warmer than observations, likely reflecting the extensive ocean area covered by sea-ice and/or the lack of local deepwater formation mechanisms like wintertime coastal polynyas in our simplified model. South of 35° S model temperature fall at the warm end of the observed data.



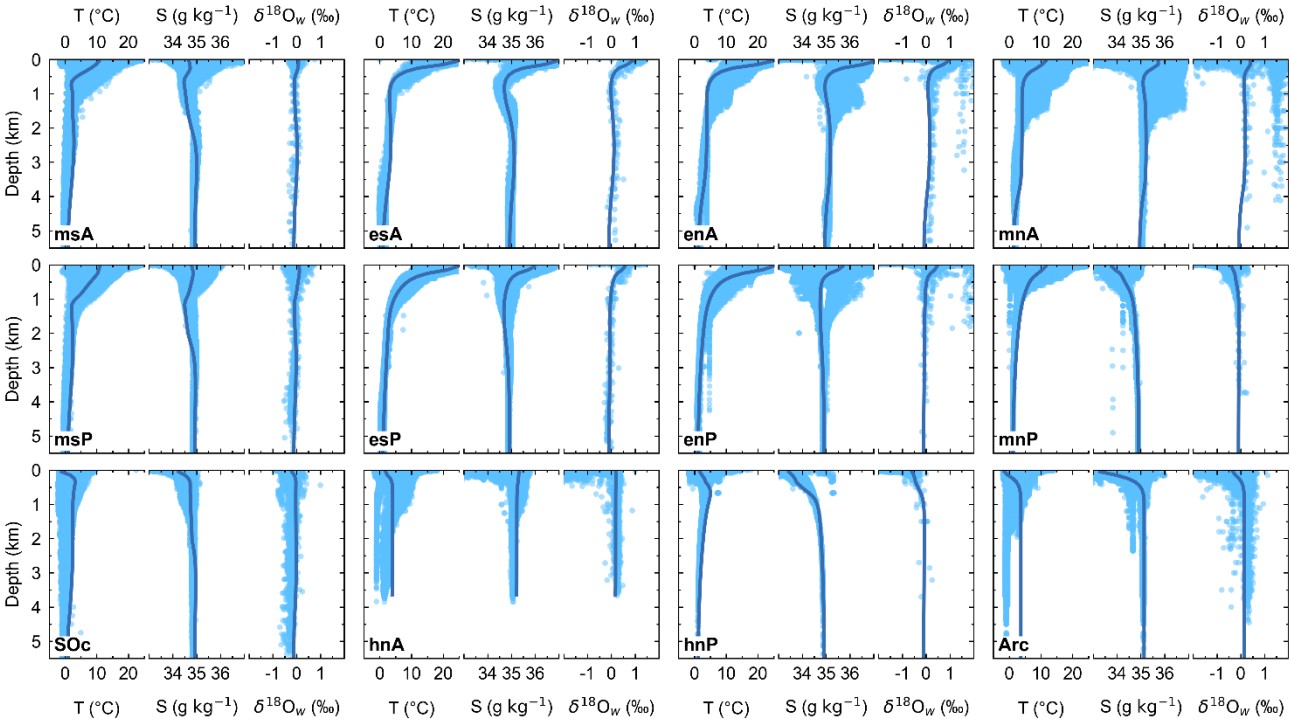

**Figure 4.** Pre-industrial, steady state model ocean vertical profiles (dark blue lines) of conservative temperature (T), absolute salinity (S) and water $^{18}$O isotopic excursion ($\delta^{18}O_w$) compared to observed data (light blue dots) for each ocean model box shown in Fig. 1a. T and S are compared with data from the GLODAPv2 database (Lauvset et al., 2022), $\delta^{18}O_w$ values are compared with data from Global Seawater Oxygen-18 Database – v1.22 (Schmidt, 1999; Bigg and Rohling, 2000, https://data.giss.nasa.gov/o18data/).

Model salinity is quite well represented and captures prominent features of global ocean, such as the salinity minimum at 650 m and 1050 m depth for msA and msP boxes representing the AAIW, the relatively surface minimum in mnP and hnP sectors associate with North Pacific Intermediate Water (NPIW; Sverdrup et al., 1942), and the vertical salinity structure in the Arctic Ocean with a surface salinity minimum, associated with freshwater inputs from runoff and sea-ice and snow melt and the strong halocline as found in observations (Aagaard et al., 1981). The lack of equatorial upwelling in our simplified model may explain the relatively warm and salty waters in the north and south tropical sectors. For $\delta^{18}O_w$, the model captures well the vertical distribution as well as the meridional gradient in each model sector, reflecting the global evaporation/precipitation distribution associated with isotopic fractionation of water.

Model ocean profiles of PO$_4$ and O$_2$ are plotted together with observational data in Fig. 5. The model achieves a good fit to the observed values in almost all ocean model boxes although the model results for O$_2$ are slightly lower than observed for the mid-depth Atlantic Ocean. Surface phosphate is strongly controlled by ocean new production. Low surface values are found



in all ocean boxes, except at the Southern Ocean where the highest surface values are found due to intense upwelling and to the low biological-production efficiency as expected from the low value of $L_f$ in our model. Furthermore, the model captures well differences between Pacific and Atlantic Ocean basins. These differences are largely a consequence of the large-scale

ocean circulation as described above. The oldest waters are found at the North Pacific Ocean (model sectors mnP and hnP, see $\Delta^{14}C$ model profiles in Fig. 6), and consequently, the highest (lowest) values of phosphate (dissolved oxygen) in the ocean interior are found there. High dissolved $O_2$ values at both surface and abyssal ocean are relatively well represented in the model in response to air-sea gas exchange and AABW ventilation. At the ocean interior, $O_2$ distributions respond mainly to organic matter remineralization and, for msA and msP sector for example, ventilation from AAIW. Misrepresentation of $PO_4$ and

dissolved $O_2$ above 1000 m depth of hnP sector is related to the way NPIW is treated in the model. The $PO_4$ (dissolved $O_2$) excess (deficit) in Arctic model sector is related to the overestimation in ocean temperature there which influences organic matter remineralization.

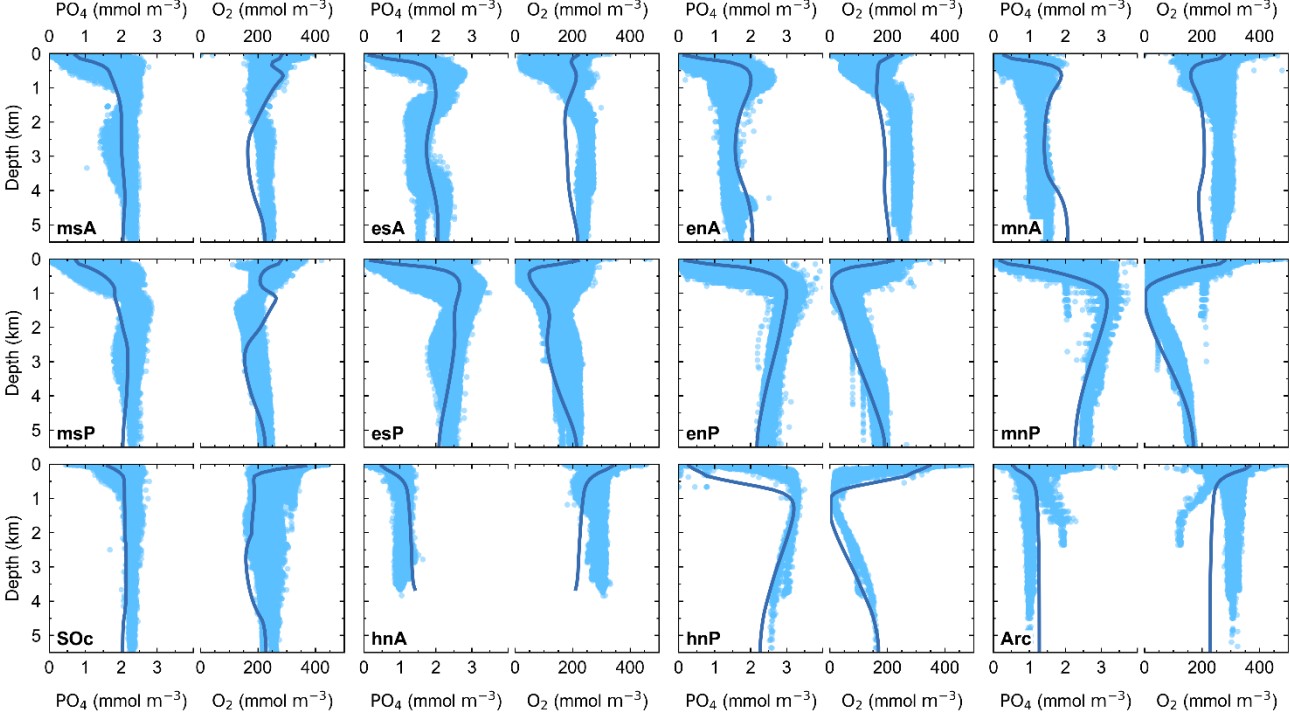

**Figure 5.** Pre-industrial, steady state model ocean vertical profiles (dark blue lines) of phosphate ($PO_4$) and dissolved oxygen ($O_2$) compared to observed data (light blue dots) for each ocean model box shown in Fig. 1a. Observed values are from the GLODAPv2 database (Lauvset et al., 2022).





The ocean carbon cycle model tracers DIC and ALK, are plotted together with observed DIC and ALK in Fig. 6. The

comparison with observational data shows excellent model-data agreement for both DIC and ALK. Both the vertical structure and the spatial differences among ocean model boxes are well captured by the model. Furthermore, both the depths of the DIC maximum and the slightly shallower depths of the $PO_4$ maximum (Fig. 5) in well-stratified ocean zones agree well with observations indicating good e-folding length choices used for model remineralization rates. These distributions reflect the interplay between the cycling of organic matter and calcite as well as the air-sea exchange of $CO_2$.


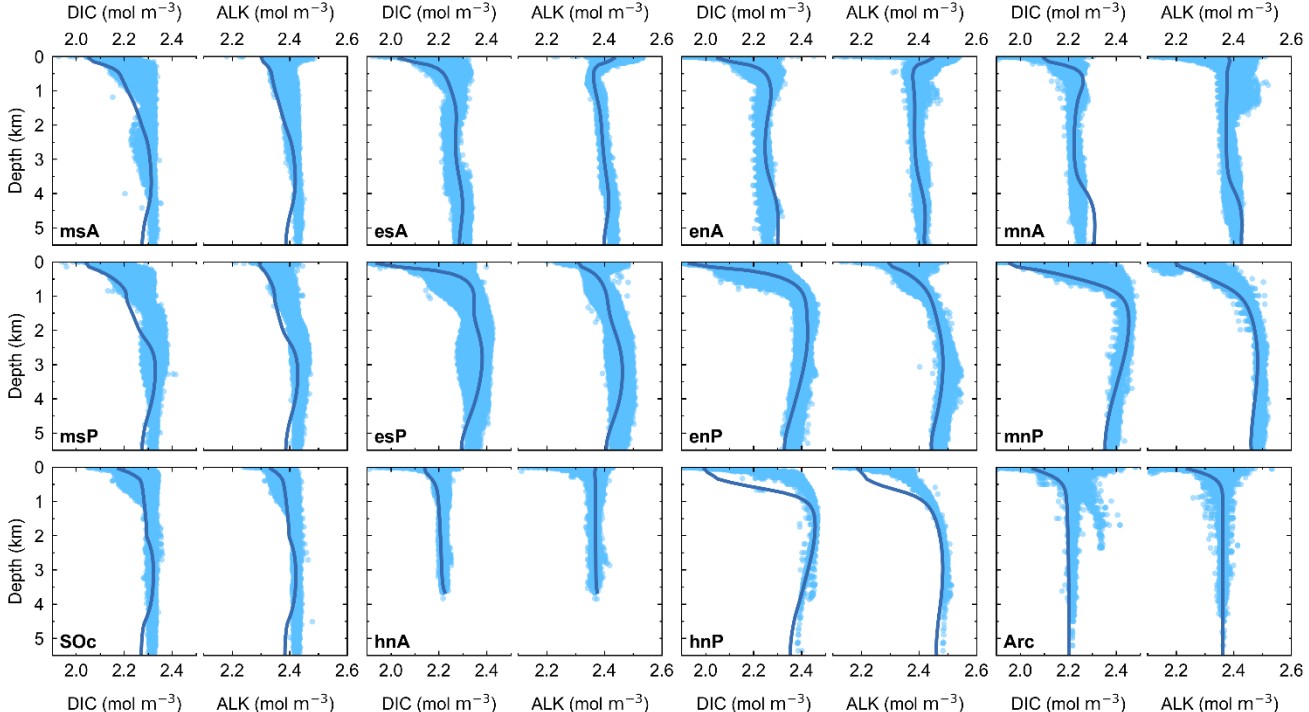

**Figure 6.** Pre-industrial, steady state model ocean vertical profiles (dark blue lines) of total dissolved inorganic carbon (DIC) and alkalinity (ALK) compared to observed data (light blue dots) for each ocean model box shown in Fig. 1a. Observed values are from the GLODAPv2 database (Lauvset et al., 2022).


Vertical distributions of carbonate ion ($CO_3^{2-}$), $CO_3^{2-}$ saturation with calcite ($CO_{3,sat}^{2-}$) and water $CO_2$ are plotted in Fig. 7. The crossing point between $CO_3^{2-}$ and $CO_{3,sat}^{2-}$ is the calcite saturation depth (CSD) which is found to be around 4000 m depth for the entire Atlantic Ocean and which reaches a minimum depth of around 700 m in the North Pacific sector where the maximum water $CO_2$ concentration even exceeds that of $CO_3^{2-}$. These results showing undersaturation throughout most of the deep North

Pacific Ocean would imply low values of $CaCO_3$ in ocean sediments there. This is confirmed by the results for modelled and observed ocean sediments presented below.



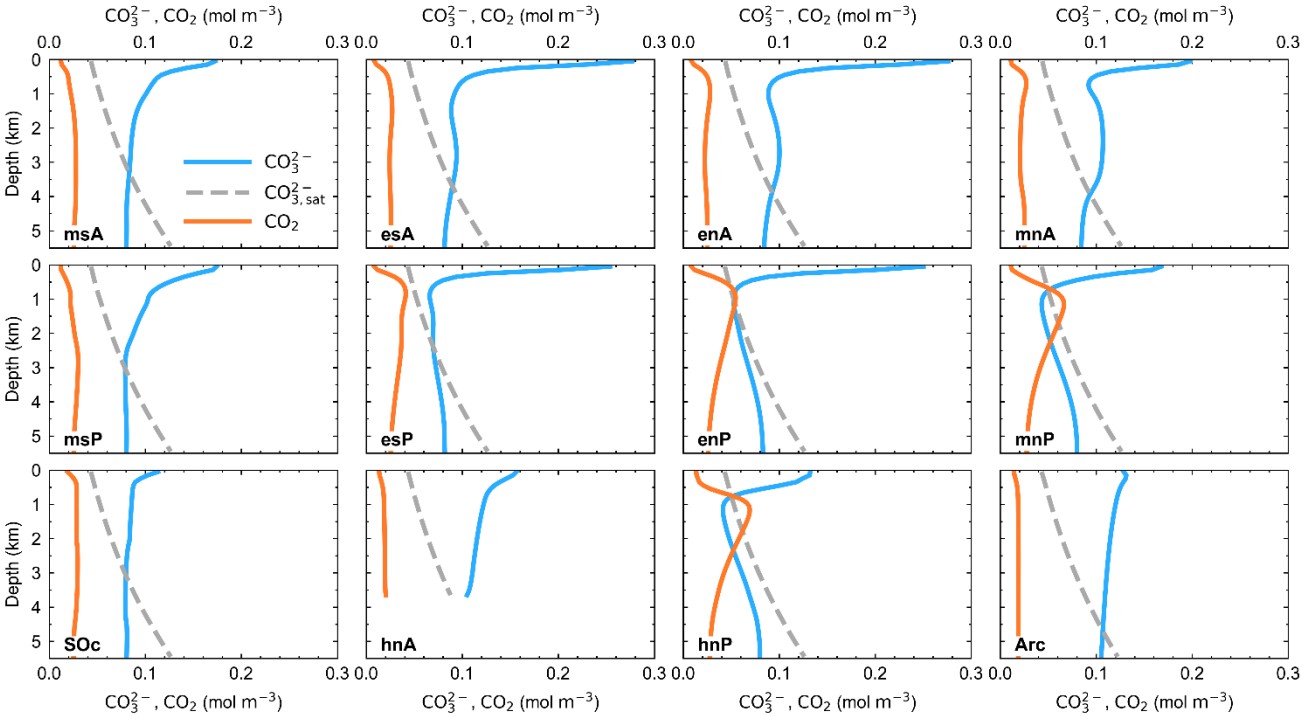

**Figure 7.** Pre-industrial, steady state model ocean vertical profiles of carbonate ion ($CO_3^{2-}$), carbonate ion saturation with calcite ($CO_{3,sat}^{2-}$), and water $CO_2$ for each ocean model box shown in Fig. 1a.


Ocean carbon isotopes ($\delta^{13}C$ and $\Delta^{14}C$) are plotted together with observational data in Fig. 8. As for DIC and ALK, there is excellent model-data agreement in nearly all model ocean zones. As light carbon is preferentially taken up during photosynthesis, export of organic matter leaves the euphotic zone enriched in $^{13}C$ (higher values of $\delta^{13}C$). Upon remineralization of this organic matter at depth, light carbon is released into the water column leaving the water there depleted

in $^{13}C$ (lower values of $\delta^{13}C$). As shown by the agreement of model $\delta^{13}C$ results with ocean as well as atmosphere data, the model deals well with this fractionation process as well as with the temperature-dependent fractionation associated with air-sea exchange of $CO_2$. As for $PO_4$ and $O_2$, meridional gradients and Pacific-Atlantic Ocean differences reflect ocean circulation as seen clearly in the $\Delta^{14}C$ profiles. The oldest waters are found in the North Pacific and exhibit the most negative values of $\delta^{13}C$ and $\Delta^{14}C$ in the ocean interior due to a longer time for receiving light carbon from organic matter remineralization and

due to a longer time for radioactive decay to act, respectively. Modelled surface ocean $\delta^{13}C$ values are well within the data-based estimates with a global surface mean model result of 2.15 ‰. As before, the model-data disagreement of $\delta^{13}C$ in the upper 1000 m depth in hnP sector is related to NPIW formation in the model.



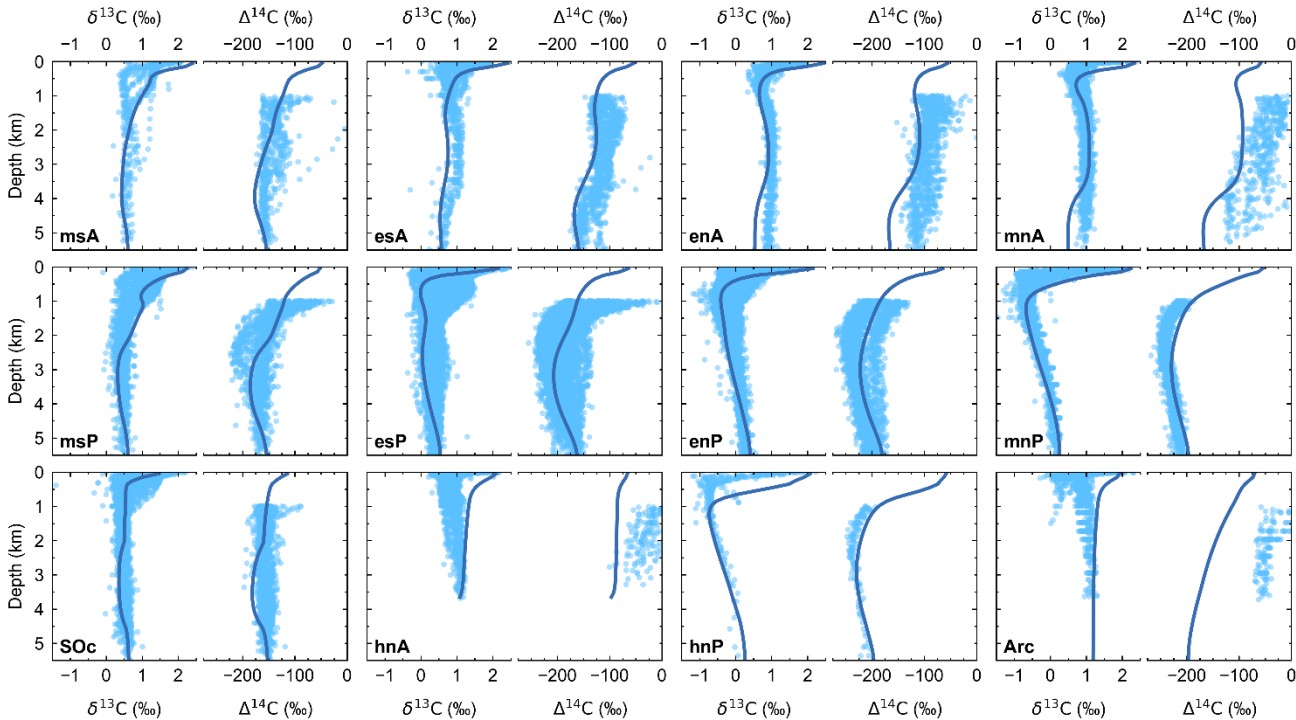

**Figure 8.** Pre-industrial, steady state model ocean vertical profiles (dark blue lines) of isotopic excursion of $^{13}$C ($\delta^{13}$C) and isotopic excursion of $^{14}$C ($\Delta^{14}$C) compared to observed data (light blue dots) for each ocean model box shown in Fig. 1a. Observed values are from the GLODAPv2 database (Lauvset et al., 2022). For $\Delta^{14}$C only values below 1000 m depth have been plotted as shallower depths are strongly affected by the anthropogenic signal including atomic bomb $^{14}$C inputs. Note that bomb $^{14}$C also affects deeper levels in the North Atlantic and Arctic boxes (mnA, hnA and Arc).

With all the above, and despite minor model-data disagreements, the model reproduces quite well the main global features of both physical and biogeochemical tracers. Better ocean modelled distributions of S, $\delta^{18}$O$_w$, DIC, ALK, $\delta^{13}$C and $\Delta^{14}$C are key improvements that have been made in comparison to the much-simpler DCESS I model where distributions of those tracers had shortcomings, in particular for S, $\delta^{18}$O$_w$ and $\delta^{13}$C (Shaffer et al., 2008).

Model global ocean new (exported) production (NP) is 5.84 Gt C yr$^{-1}$, slightly higher than the 5.4 Gt C yr$^{-1}$ of the DCESS I model (Shaffer et al., 2008). This new estimate is at the lower end of the estimates from Dunne et al., (2007) (9.6±3.6 Gt C yr$^{-1}$) but almost matches a modern and possibly more robust estimate of Siegel et al., (2014) of 5.90 Gt C yr$^{-1}$. By far, the tropical ocean model zones (35° S-35° N) are the most productive model regions, accounting for more than 66 % of global exported production partitioned into a 53 % and 13 % for the Pacific and Atlantic Ocean respectively. The Arctic Ocean and high north latitude zone of Pacific Ocean have the lowest values of exported NP with 0.2 % and 1.1 % of the total respectively. The Southern Ocean also has a rather low value with 5 % of total export production. Northern and southern mid-latitude model





sectors (35°-55° N/S) show around 22 % of total NP, with 5 % and 7 % for mnP and mnA and 8 % and 2 % for msP and msA boxes respectively. This meridional distribution is also observed in the publications cited above. We believe that lack of

equatorial upwelling of rich-nutrient waters in the model probably contributed to the relatively low value of ocean NP in our model.

Global biogenic calcite production in the model is 1.25 Gt C yr⁻¹ where more than 65 % is produced in tropical model zones of the Pacific Ocean. Tropical zones of the Atlantic Ocean account only for 16 % of global calcium carbonate production.

Extratropical ocean model zones from Northern and Southern hemispheres and combining Pacific and Atlantic boxes represent only a 15 % of total biogenic calcite production. Table 4 shows values for organic matter and calcite production for each ocean model box.

**Table 4: Pre-industrial organic matter and biogenic calcite ocean production estimates. All values are given in Gt C yr⁻¹.**

| Ocean limits | Atlantic Ocean | | Pacific Ocean | |
|---|---|---|---|---|
| | Organic matter | Biogenic Calcite | Organic matter | Biogenic Calcite |
| 65° N-55° N | 0.20 | 0.01 | 0.07 | $3.52\times10^{-3}$ |
| 55° N-35° N | 0.38 | 0.06 | 0.30 | 0.04 |
| 35° N-0° | 0.43 | 0.12 | 1.34 | 0.35 |
| 0°-35° S | 0.35 | 0.09 | 1.80 | 0.47 |
| 35° S-55° S | 0.15 | 0.02 | 0.47 | 0.07 |
| | Southern Ocean (55° S-70° S) | | Arctic Ocean (90° N-65° N) | |
| | 0.31 | 0.01 | 0.02 | $5.43\times10^{-4}$ |


Our estimates are well into the observed range of 0.5-1.6 Gt C yr⁻¹ of Berelson et al., (2007) but slightly higher than more recent estimates of 0.91±0.14 Gt C yr⁻¹ (Sulpis et al., 2021). Overall, our model result of new (exported) production is well into the range of CMIP6 models (Planchat et al., 2023), but our biogenic calcite production is at the upper end of those model estimates. However, we note that CMIP6 ensemble results tend to globally underestimate the exported inorganic carbon

compared to observational estimates (Planchat et al., 2023).

The calcite carbon to organic carbon flux ratio is similar for the Pacific/Atlantic Oceans, with ratios of 0.6/0.5, 0.9/0.7 and 1.2/1.1 in tropical model sectors at 1000, 2000 and 3000 m depth respectively, and ratios of 0.2, 0.3 and 0.5 for northern latitudes sectors of those basins at these depths. These ratios are somewhat lower than those estimated from sediment trap data

(Berelson et al., 2007). The longer e-folding scale $\xi_{cal}$ than $\xi_C$ and the temperature effect on organic matter remineralization and on the $CaCO_3$ rain ratio (see Sect. 2.4) help explain these results.





### 3.1.6 Sediment results

In the model pre-industrial steady state, 0.11 Gt C yr$^{-1}$ of organic carbon is buried in ocean sediments. This represents 12 % of total organic carbon falling on sediment surface; the remaining 88 % is remineralized to DIC that returns to the water column.

More than 90 % of this burial takes place at water depths shallower than 1000 m. The highest modelled burial rates for organic carbon per ocean sector are found in the tropical Pacific Ocean and the North Atlantic Ocean (35° N-65° N) sectors with over 40 % and 27 % of the total organic carbon burial, respectively. The global inventory of bioturbated layer (BL) organic carbon is 122 Gt C, somewhat higher than found in the pre-industrial, DCESS I model simulation (92 Gt C).

For calcite, the global annual mean burial rate is 0.21 Gt C yr$^{-1}$ where more than 57 % takes place in the tropical Pacific sectors and 23 % is buried at the combined north-tropical and mid-latitude Atlantic Ocean sectors. Of the total annual mean burial rate of $CaCO_3$ in the Pacific and Atlantic Ocean (0.14 and 0.07 Gt C yr$^{-1}$ respectively) 49 % and 45 % of calcite is buried at depths greater than 1000 m in these basins, respectively. Modelled global ocean mean calcite dry-weight fraction $(CaCO_3)_{dwf}$ is 0.349, whereas Pacific and Atlantic oceans present mean values of $(CaCO_3)_{dwf}$ of 0.353 and 0.440 respectively, where highest values

are found in south tropical Pacific and north Atlantic sectors with values of 0.459 and 0.553 respectively. Model global inventory of bioturbated layer calcite carbon is 767 Gt C, somewhat lower than the DCESS I pre-industrial estimate (1010 Gt C) but in good agreement with the 800 Gt C data-based estimate from Archer (1996). Of this, 503 Gt C is found in the Pacific Ocean BL and 249 Gt C in the Atlantic Ocean one, equivalent to 66 % and 32 % of the total, respectively. The results above highlight how the model captures ocean carbon cycle differences between Pacific and Atlantic Ocean.


Figure 9 shows the dry-weight fractions of organic matter $(OM)_{dwf}$ and calcite $(CaCO_3)_{dwf}$ as well as the sediment velocity out of the bottom of the BL ($w_{sed}$) for each ocean model box. In general, relatively high $(OM)_{dwf}$, relatively low $(CaCO_3)_{dwf}$ and rapid sedimentation rates are found at shallow depths. This can be explained mainly by high prescribed flux of non-calcite minerals at such depths that 1. fill the BL mainly with such minerals and 2. rapidly flush the BL thereby promoting high

$(OM)_{dwf}$ burial since relatively little organic matter is remineralized in the BL on such short time scales. At intermediate depths above the model's CSD, $(CaCO_3)_{dwf}$, is higher, $(OM)_{dwf}$ is lower and sedimentation rates are more moderate. This results from enhanced $CaCO_3$ accumulation in sediments bounded by ocean layers supersaturated with carbonate ion (see Fig. 7), but also, from much lower non-calcite mineral and organic matter rain rates leading to slower BL flushing that allows more complete organic matter remineralization. At those intermediate depths, the modelled effect of dissolved oxygen on organic matter

remineralization can be seen in the North Pacific model sectors where this remineralization is slowed down considerably as ocean layers there become nearly suboxic/anoxic. This also leads to a local increase in sediment rates. In the deep ocean below the model's CSD, rapid decreases in $CaCO_3$ content, very low organic matter contents and slower sedimentation rates are the result of calcite dissolution combined with low constant non-calcite rain rates and still lower organic matter rain rates. The Arctic Ocean model sediment is almost completely composed of non-calcite mineral as the sediment flux of biogenic material



falling on sediment surface is very small due to the extremely limited surface new production. In the Southern Ocean, model

sediment is also composed mainly of non-calcite-mineral but contains larger fractions of $(OM)_{dwf}$ and $(CaCO_3)_{dwf}$ than in the

Arctic Ocean. The water depth of the calcite compensation depth (CCD), defined operationally where $(CaCO_3)_{dwf}$ equals 0.1,

is found around 4800 m for both tropical Pacific and tropical Atlantic Ocean sectors. In contrast, there are major differences

between North Atlantic and North Pacific sectors with CCD depths around 4400 m and 2100 m depth, respectively.


**Figure 9.** Pre-industrial, steady state model sediment vertical profiles of dry-weight fractions of organic matter, $(OM)_{dwf}$, and calcite, $(CaCO_3)_{dwf}$, as well as sedimentation velocity $(w_{sed})$, all at the base of the bioturbated layer of each ocean model box shown in Fig. 1a.

**3.1.7 Lithosphere results**

Table 5 lists model weathering and burial rates for the pre-industrial steady state. From weathering-burial steady-state balances

outlined in Sect. 2.7 and global mean values of $^{13}C$ isotope excursions for organic carbon ($\delta^{13}OrgC$ = -22.06 ‰) and carbonate

($\delta^{13}Carb$ = 2.22 ‰) obtained from the fifth calibration step, modelled outgassing from volcanoes is found to be 0.140 Gt C yr

$^{-1}$, in good agreement with to modern-day data-based estimates (Masson-Delmotte et al., 2021).






**Table 5: Weathering, lithosphere outgassing and burial estimates for pre-industrial, steady state balances.**

| Property | Symbol | Estimated value |
|---|---|---|
| Phosphorus weathering | $W_{P,PI}$ | $2.507 \times 10^3$ mol P s$^{-1}$ |
| Organic carbon weathering | $W_{OrgC,PI}$ | 0.064 Gt C yr$^{-1}$ |
| Carbonate weathering | $W_{Cal,PI}$ | 0.115 Gt C yr$^{-1}$ |
| Silicate weathering | $W_{Sil,PI}$ | 0.098 Gt C yr$^{-1}$ |
| Lithosphere outgassing | $Vol_{PI}$ | 0.140 Gt C yr$^{-1}$ |
| Organic carbon burial | $B_{OrgC,PI}$ | 0.106 Gt C yr$^{-1}$ |
| Carbonate burial | $B_{Cal,PI}$ | 0.213 Gt C yr$^{-1}$ |

The modelled river input of total inorganic carbon is 0.43 Gt C yr$^{-1}$, a value close to that found in the DCESS I model (0.40 Gt C yr$^{-1}$) and close to the 0.41 Gt C yr$^{-1}$ from data-based estimates (Li et al., 2017). More than 51 % of this river input flux, is

delivered to the Atlantic Ocean, while 40 % drains to the Pacific Ocean and the rest to the Arctic Ocean. The Southern Ocean does not receive inorganic carbon from rivers. With the results above, modelled steady state pre-industrial global ocean outgassing of CO$_2$ is 0.11 Gt C yr$^{-1}$ which is balanced by net uptake of atmospheric CO$_2$ that is expressed as $2W_{Sil,PI} + W_{Cal,PI} - (Vol_{PI} + W_{OrgC,PI})$.

**3.2 Testing the model**

We carry out two simple experiments to test model performance. In the first experiment (Exp 1) we force the model with an imposed, time-dependent function of northward Ekman transport out of the Southern Ocean at 55° S. We considered two cases for this experiment, doubling (Exp 1a) and halving (Exp 1b) this transport value on a timescale of 1500 years. With this, we try to emulate the role of Southern Ocean westerly winds on large-scale ocean circulation as it has been hypothesized that intensity and position shifts of these winds may drive global-scale climate changes on glacial-interglacial timescales (Gray et

al., 2023; Lee et al., 2011; Toggweiler et al., 2006). In the second experiment (Exp 2) we force the model with an imposed, time-dependent function of freshwater input to the Southern Ocean shelf to emulate a melting pulse of the Antarctic ice sheet. We "melt" the equivalent of $2.5 \times 10^{15}$ m$^{-3}$ freshwater (equivalent to 5 m sea level rise) over two different timescales: a fast pulse to get a sea level rise of 1 m (Exp 2a) and a slow pulse to get a 0.1 m rise (Exp 2b) after 250 years of model simulation. This is consistent with modelled assessments about Antarctic ice sheet retreats in the present climate context (Ruckert et al.,

2017; Deconto and Pollard, 2016). In both experiments we focus on changes on large scale ocean circulation and its impact on ocean tracer distributions as well as on global climate.





### 3.2.1 Southern Ocean Ekman transport forcing

The forcing and model results are presented in Fig. 10. With the increase of northward Ekman transport (Exp 1a) across 55°
S, there is more upwelling in the Southern Ocean sector (Fig. 10b). However, this Southern Hemisphere forcing does not

significantly impact AMOC intensity in the North Atlantic Ocean, as shown by the streamline evolution at 55° N at 1000 m
depth (Fig. 10e). Rather, this forcing leads to the appearance of a clockwise recirculation cell between 55° S and 40° S confined
between 1000 m and 2000 m depth for both Atlantic and Pacific Oceans, as indicated by the vertical flux (defined positive
upwards) in msA and msP sectors at 1500 m depth showed in Fig. 10c-d. In Exp 1b, a decrease in northward Ekman transport
across 55° S causes a decrease in the Southern Ocean upwelling (Fig. 10b). In this experiment there is an increase in the upward

flux above 2000 m depth in msA and a decrease in the downward flux below 1000 m depth in msP, fluxes that subsequently
join the upper ocean northward flow and the deep southward flow in the Atlantic and Pacific Oceans respectively. Insensitivity
of North Atlantic AMOC intensity as well as intermediate-depth Southern Hemisphere recirculation for increased westerlies
forcing has also been reported in other modelled experiments (Jochum and Eden, 2015; Rahmstorf and England, 1997).



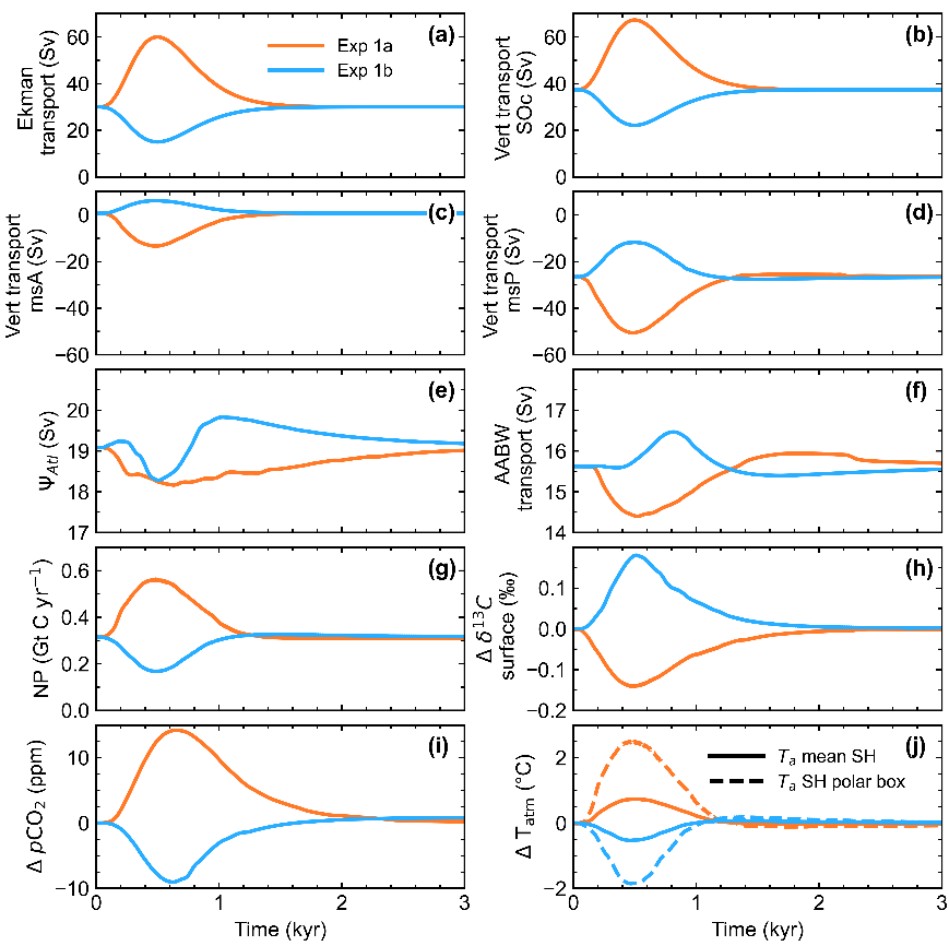


**Figure 10.** Results for 3000-year simulations for two different choices of northward Ekman transport across 55° S. **(a)** Applied Ekman transport evolutions, **(b)-(d)** vertical transport (Vert transport, defined positive upwards) at 1500 m depth for Southern Ocean, msA and msP sectors respectively, **(e)** northward transport across 55° N above 1000 m depth in the Atlantic Ocean, **(f)** AABW transport reaching the abyssal Southern Ocean, **(g)-(h)** Southern Ocean new production evolution and Southern Ocean surface $\delta^{13}C$ change and **(i)-(j)** changes in atmospheric $CO_2$ and atmospheric temperature (solid lines are for Southern Hemisphere mean $T_a$ and dashed lines are for $T_a$ for atmospheric southern polar zone, 55° S-90° S).


These ocean interior circulation changes lead to a denser (lighter) water being upwelled in the Southern Ocean for Exp1 and Exp 2 respectively, leading to less (more) entrainment and consequently decreasing (increasing) amounts of AABW reaching

the abyssal ocean for these experiments (Fig. 10f). More (less) upwelling together with this Southern Hemisphere recirculation, enhance (reduce) ocean new production in the Southern Ocean, as more (less) nutrients are carried to the surface there (Fig. 10g). These changes in Southern Ocean NP are not reflected in surface ocean $\delta^{13}C$ values (Fig. 10h, as well as in msA and msP) in this zone of low model new production efficiency. Rather, the decrease (increase) in $\delta^{13}C$ values in Exp 1a (Exp 1b) is associated with lighter (heavier) signal of $^{13}C$ from more (less) upwelling of deeper water originating from northern latitudes





of Atlantic and Pacific model zones. Air-sea exchange of $CO_2$ with the Southern Ocean reflects changes in the upwelling rates there as more or less carbon-rich deeper water is carried to surface layers there. More (less) outgassing there leads to an increase (decrease) in atmospheric $CO_2$ concentration (Fig. 10i) and a resulting rise (fall) in atmospheric temperature (Fig. 10j). Changes in atmospheric $CO_2$ content is nearly uniform in all atmospheric boxes due to the rapid turnover time of the atmosphere. The temperature change in southern polar atmospheric box is around 2.5 °C and -1.8 °C for Exp 1a and Exp 1b respectively. However, this warming (cooling) signal decreases rapidly farther northward such that mean Southern Hemisphere atmospheric temperature changes are around 0.7 and -0.5 °C for Exp 1a and 1b respectively.

### 3.2.2 Antarctic ice sheet melt freshwater forcing

Forcing and model results for this experiment are shown in Fig. 11. The rapid freshwater pulse to the Southern Ocean shelf (Exp 2a) produces a greater and more abrupt decrease in AABW fluxes flowing down off the shelf break ($F_{sl,0}$), as well as in the abyssal ocean, $F_{sl}$, (Fig. 11b). For Exp 2a, $F_{sl,0}$ and $F_{sl}$ reach minimum values of 3.5 and 11 Sv respectively, i.e. around 30 % less than the pre-industrial values. In Exp 2b, modest declines of 0.7 and 2 Sv are obtained for $F_{sl,0}$ and $F_{sl}$, representing a decrease of only 13 % compared to modern-day values (Gordon, 2019). Consequently, the abyssal counterclockwise circulation decreases its intensity by more than 11 and 2 Sv for fast and slow freshwater pulses respectively, as shown by the northward streamline evolution at 4500 m depth crossing 55° S (Fig. 11c). This corresponds to a 75 % decrease compared to only a 14 % decrease for the fast and slow freshwater pulses, respectively, underlining the sensitivity of model AABW circulation to the speed and not only the size of the Antarctic Ice Sheet melt water pulse. Such lower AABW ventilation rates lead to the decline in dissolved $O_2$ content of the abyssal northern sectors of the Pacific and Atlantic Ocean (Fig. 11d-e). A greater AMOC penetration depth, as a consequence of weakened AABW abyssal circulation, may inhibit a further decrease in the abyssal $O_2$ content of the North Atlantic Ocean as more atmospheric oxygen is carried to those depths.

Opposite behaviors are observed in the abyssal ocean $\Delta^{14}C$ content for northernmost model sectors of North Atlantic and North Pacific Ocean (hnA and hnP, respectively). The increase in the hnA sector (Fig. 11f) is caused by deeper AMOC penetration depth as AABW cell gets weaker leading to increased transport of near surface $^{14}C$ to the abyssal ocean. In the hnP sector, the decrease in $^{14}C$ content stems from longer residence times due to the reduced AABW ventilation there leading to more radioactive decay. The oxygen-18 signature in the abyssal ocean is quite sensitive to the freshwater forcing to the Southern Ocean shelf as water melt from the Antarctica ice sheet comes with an isotopic signal of -40 ‰ in the model. This is shown in Fig. 11 h-i, where similar changes are observed at the boxes directly north of the Southern Ocean (msA and msP) at 4500 m depth, with changes around of -0.17 and -0.09 per mil for fast and slow forcing experiments, respectively. Despite all these ocean interior changes, Exp 2a and Exp 2b forcings lead only to small changes in atmospheric $CO_2$ of around 1 ppm or less in both experiments. As a consequence, there is nearly no change in global atmospheric temperature although small, sea-ice dynamics-driven oscillations of 20 year-period are found in southern polar atmospheric temperature (not shown). We stress

that this simplified experiment was carried out mainly to test the model performance to isolated fresh water forcing. A comprehensive study of the global climate system response to Antarctic ice sheet melting would also require consideration of

climatic effects leading to the melting in the first place, i.e. a coupling of the climate system and the ice sheet.

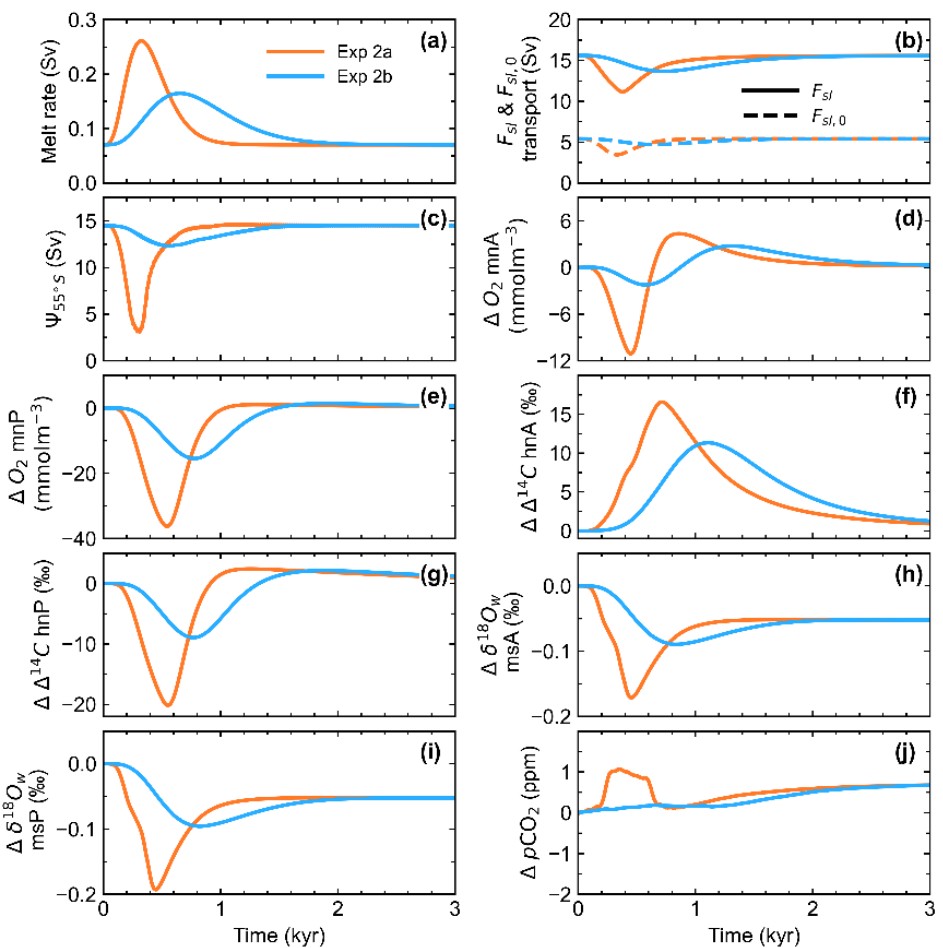

**Figure 11.** Results for 3000-year simulations for two different choices of Antarctic icesheet melt rate. **(a)** Applied melt rates evolutions, **(b)** AABW transport flowing off the Southern Ocean shelf ($F_{sl,0}$) and reaching abyssal depths ($F_{sl}$), **(c)** total northward transport (combined
Atlantic and Pacific) crossing 55° S below 4500 m depth, **(d)**-**(e)** changes in dissolved $O_2$ content in the ocean at 4500 m depth for northern sectors of the Atlantic and the Pacific Ocean, respectively (mnA and mnP), **(f)**-**(g)** ocean $\Delta^{14}C$ change at 4500 m depth for the northernmost sectors of the Atlantic and the Pacific Ocean, respectively (55° N-65° N, hnA and hnP), **(h)**-**(i)** $\delta^{18}O_w$ change at 4500 m depth for ocean boxes directly north of the Southern Ocean (msA and msP) and **(j)** global mean atmospheric $pCO_2$ change.

## 4 Discussion and conclusions

We have described and tested a new Earth System Model of Intermediate Complexity, DCESS II, that has been calibrated to

the pre-industrial Earth System. For this relatively simple model, efforts were made to limit the number of model free


parameters and to constrain their values as much as possible using observations. At the end of this process, we find generally excellent agreement with available modern observations.

Some key features of the physical part of the model like horizontal resolution, a simplified dynamical scheme for large-scale ocean circulation, stratification-dependent vertical diffusion, a gravity current approach to the formation of Antarctic Bottom Water as well as a dynamical formulation for sea-ice, have helped us to achieve good agreement to observed data like atmospheric heat and water vapor transport and ocean temperature and salinity distributions (Lauvset et al., 2022). These improvements with respect to the original DCESS I model, together with updates in ocean biogeochemistry such as a light-

dependence on surface new (exported) production of organic matter, dependence on calcite saturation state for the biogenic carbonate production, as well as local temperature dependence on organic matter remineralization, result in accurate model representations of ocean biogeochemical tracers, highlighted by an excellent model-data agreement with regard to marine carbon cycle species. Furthermore, the incorporation of a new land biosphere module considering three different types of vegetation and a light-atmospheric $CO_2$-dependence on net primary production on land, as well as ocean sediment and

lithosphere modules applied to multiple ocean and land sectors make the DCESS II model an excellent while economical tool for studying and gaining short as well as long-term understanding of the global carbon cycle.

Two experiments were conducted in order to explore and test model performance. In the first experiment (Exp 1) we forced the model in a way that emulates the role of the Southern Hemisphere westerly winds on the large-scale ocean circulation. In

the second experiment (Exp 2) we introduced (at two different rates) a freshwater "melt" volume equivalent to a global sea level rise of 5 m into the shelf surrounding Antarctica. In both experiments, model results show important physical and biogeochemical changes in the ocean driving perturbations in the global carbon cycle. This is reflected by responses in atmospheric $CO_2$ as well as in abyssal ocean $\Delta^{14}C$ and dissolved $O_2$ content. These experiments serve to demonstrate how the model captures the global role that Southern Hemisphere westerly winds and Antarctic Bottom Water play in the Earth System.


The development and calibration of the DCESS II model as described here has also set the stage for future work. For example, coupling of the model to an Antarctic Ice Sheet (AIS) model will provide a more realistic Earth System simulation when compared to our idealized experiment 2 described above. For this work an extended and improved version of the simple, well tested DAIS model (Shaffer, 2014) will be used. In another example, planned incorporation of methane and nitrogen cycles

into DCESS II will significantly improve its ability to deal realistically with deep-time global warming events associated with massive carbon inputs to the Earth System. Groundwork for this step has already been done in as much as these global biogeochemical cycles were incorporated into the DCESS I model (Shaffer et al., 2017). In addition, other potential model applications include the study of different global climate events like Dansgaard-Oeschger oscillations, longer climatic events like the Eocene-Oligocene Transition or even the assessment of deep-time mass extinction events by taken advantage of the



relative simplicity of the model for setting proper boundary conditions like orbital forcing and continental distribution among others.

In conclusion, we have presented, validated and tested a simple and fast new Earth System Model of intermediate complexity intended to be a flexible, comprehensive and economical Earth System modelling platform. Despite its limitations like

relatively low horizontal resolution and multiple parameterizations, the model represents quite well most Earth System components, especially the global-scale carbon cycle. Due to its simplicity, it could be easily modified in terms of boundary conditions to address specific past epochs. Thus, we find DCESS II to be a useful tool for studies of past, present and future global change on time scales of years to millions of years while not needing large computational resources.

*Code availability.* The code for the DCESS II model and others DCESS model versions are freely available at

http://www.dcess.dk/.

*Data availability.* Ocean observed data used in Figs. 4-6 and in Fig. 8 are from the Global Ocean Data Analysis Project version 2.2022 (GLODAP-v2.2022, Lauvset et al., 2022) database and can be accessed at the National Oceanic and Atmospheric Administration (NOAA) National Centers for Environmental Information (NCEI) under https://doi.org/10.25921/1f4w-0t92. Observed values of seawater $^{18}$O showed in Fig. 4 are from the Global Seawater Oxygen-18 Database - v1.22 and can be

accessed at https://data.giss.nasa.gov/o18data/ (Schmidt, 1999; Bigg and Rohling, 1999). Sea-ice observations are from the Sea Ice Index, Version 3 of the National Snow and Ice Data Center, Boulder, Colorado, USA (Fetterer et al., 2017) and can be accessed at https://doi.org/10.7265/N5K072F8. World Ocean Atlas database (Boyer et al., 2018) used for calculation of reference temperature for organic matter remineralization can be accessed at NOAA NCEI under https://www.ncei.noaa.gov/archive/accession/NCEI-WOA18. Southern Hemisphere atmospheric $CO_2$ data were obtained

from the Cape Grim Baseline Air Pollution Station belonging to the Commonwealth Scientific and Industrial Research Organisation (CSIRO) Oceans & Atmosphere and the Australian Bureau of Meteorology. This data can be freely downloaded at https://capegrim.csiro.au/. Northern Hemisphere atmospheric $CO_2$ data were obtained from the Mauna Loa Observatory belonging to NOAA Earth System Research Laboratory (ESRL). This data can be freely downloaded at https://gml.noaa.gov/ccgg/trends/. Present-day volcano distributions were obtained from the Volcano Location Database

belonging to the NOAA NCEI. These data can be freely downloaded from https://www.ngdc.noaa.gov/hazel/view/hazards/volcano/loc-data. Reference wind values for air-sea exchange are from the NOAA/CIRES/DOE 20th Century Reanalysis (V3) dataset provided by the NOAA PSL, Boulder, Colorado, USA, from their website at https://psl.noaa.gov.

*Author contributions.* EF and GS designed the work, EF led the development of the new model and code to which GS

contributed. EF wrote the original draft, and together with GS discussed model concepts and results and wrote the final manuscript.

*Competing interests.* The authors declare that they have no conflict of interest.

*Acknowledgements.* We thank to the Chilean National Agency for Research and Development (ANID) for financial support of this work which was largely supported through the ANID/CONICYT-PFCHA/Doctorado Nacional/2017-21171747 and



FONDECYT (Chile) 1190230 grants. We also received support from FONDECYT (Chile) grant 1230534 and the ANID Millennium Science Initiative Program (Millennium Institute of Oceanography ICN12_019). In addition, we thank Roberto Rondanelli who helped us with equatorial-cross heat transport parameterization. Finally, we would like to extend special thanks to Donald Canfield who invited EF to spend time at his lab at the University of Southern Denmark and whose valuable comments and discussions helped us with the formulation of the ocean biogeochemistry module.

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
