# Peer review of "Presentation, Calibration and Testing of the DCESS II Earth System Model of Intermediate Complexity (version 1.0)"

_Geoscientific Model Development, 2024_

## Author Response (AR1)

**Reply to referee #1**

We thank referee #1 for the careful review of our paper and the suggestions for further enhancement. We are very pleased in the referee´s confidence that our new model "will be valuable to the biogeochemistry community and beyond" and "has great potential for many new fascinating applications". In the following we address the referee´s comments and indicate changes/additions made to our revised manuscript in response to the comments.

Below, the referee's comments are in black, our answers are in blue and text changes are in blue italics within quotation marks.

Only new references included in our answers are listed at the end of this document.

**Major comments:**

I agree that increasing model complexity is one direction for development; however, it may also introduce indefinite parameters and uncertainties and undermines the generality of the model. In the current manuscript, the motivation for making models more complex is lacking. It would be better to add a discussion on what the limitations of the original model were, why the model needs to be made more complex, and in what ways the new model has yielded better results than the original model.

We agree with the referee that the discussion suggested would improve our manuscript. We now include an enhanced discussion of the limitations of the original model and the motivations to make a new model in the second paragraph of section 1. The revised, second half of this paragraph now reads:

"… . *The DCESS I model has proven to be useful tool in such studies as documented by many stand-alone or intercomparison study publications (e.g. Eby et al., 2013; Harper et al., 2020; Joos et al., 2013; Macdougall et al., 2020; Shaffer, 2010; Shaffer and Lambert, 2018; Shaffer et al., 2009; Zickfeld et al., 2013). Nonetheless, DCESS I has serious limitations for addressing many important Earth System problems, like glacial-interglacial cycles for example, due to its one-hemisphere, two-sector horizontal resolution, lack of ocean dynamics, lack of seasonal cycles and simplified land vegetation scheme, among other factors. In order to address those deficiencies, here we present a new, Earth System Model, DCESS II, that contains great improvements in model geometry and physical/biogeochemical processes. This new model is able to capture not only relevant environmental differences between major ocean basins, but also, for example, to produce synthetic ocean sediment cores from distinct ocean zones for more detailed comparison with data, and this while retaining much of the simplicity and the spirit of the DCESS I model. Thus, DCESS II is a simple, fast and highly flexible ESM of intermediate complexity well suited to run long-term experiments in a relatively simple way with no need for large computational resources.*"

Furthermore, in the second paragraph of section 4 we enhance our original discussion of the better results of DCESS II compared with DCESS I to read:

*"Some key features of the physical part of the model like horizontal resolution, a simplified dynamical scheme for large-scale ocean circulation, stratification-dependent vertical diffusion, a gravity current approach to the formation of Antarctic Bottom Water as well as a dynamical formulation for sea-ice, have helped us to achieve good agreement to observed data like atmospheric heat and water vapor transport and ocean temperature and salinity distributions (Lauvset et al., 2022) with space and time resolution not possible using the original DCESS I model. These improvements, together with updates in ocean biogeochemistry such as a light-dependence on surface new (exported) production of organic matter, dependence on calcite saturation state for the biogenic carbonate production, as well as local temperature dependence on organic matter remineralization, result in accurate model representations of ocean biogeochemical tracers, highlighted by an excellent model-data agreement with regard to marine carbon cycle species, well beyond that achieved with the DCESS I model in particular with respect to carbon 13 (Shaffer et al, 2008). Furthermore, the incorporation of a new land biosphere module with three different vegetation types and a light-atmospheric $CO_2$-dependence on net primary production on land, as well as ocean sediment and lithosphere modules applied to multiple ocean and land sectors make results of the new model much more amenable to comparison with data than is the case for the DCESS I model. All the above improvements help make the DCESS II model an excellent while economical tool for studying and gaining short as well as long-term understanding of the global carbon cycle."*

*To underline this, we also extended the last part of the first paragraph of section 4 to read:*

*"... using observations. At the end of this process, despite model limitations, we generally find excellent agreement with available modern observations. When compared with results from the DCESS I model, a number of improvements with our new model stand out as described below."*

This paper ought to do more to acknowledge the general caveat that this type of model generally contains many tunable parameters, and that model assumptions (the appropriateness of the chosen functions, parameterizations, and simplifications) are difficult to validate. Given this caveat, clear discussions about the model assumptions and primary limitations are essential. For example, the implementation of a dynamic scheme of ocean circulation is one of the key improvements of this study, but because of "regional tunings", this scheme cannot be used as is for the deep past, where the continental configuration is largely different from the present.

In the following we address this point made by the referee, but we first insist that we have made a significant effort to limit the number of tuneable parameters and where needed they were anchored to the greatest extent possible in observational results.

With regard to "regional tunings", there are only two such tunings per se in our model, one associated with polar reduction of horizontal mixing, as has been used in other work referred to in section 2.3, and a second one related to net ocean primary production efficiency in polar regions. With the first one, the model is able to reproduce global ocean intermediate waters (Antarctic Intermediate Water and North Pacific Intermediate Water); with the second one, the model is able to reproduce in a simplified manner observed, unused nutrients in polar regions due to iron and/or light limitation. Otherwise, the same horizontal mixing (and indeed the same friction coefficient governing the large-scale overturning circulation) is used throughout the model's oceans. We believe that our model can

indeed be used to great advantage for studying deep-time past climates where continental distribution is quite different from today if sufficient care is taken when considering changes in continental configuration. To underline this, we revised the end of the second-to-last paragraph in section 4 to read:

*"... for setting proper boundary conditions like orbital forcing and continental distribution among others. In this regard, when such deep-time model applications with quite different boundary conditions from today are to be made, careful consideration should be made in choosing the few (only two) regional tunings of the model."*

Additionally, the ignorance of ocean biogeochemical processes under anoxic conditions (e.g., denitrification and sulfate reduction) is not discussed in the paper. This simplification means that this model cannot evaluate the biogeochemistry in anoxic oceans. More discussion of model limitations would be helpful in preventing misunderstanding and misapplications of the model.

We are very aware of this but have opted to leave the treatment of suboxic/anoxic conditions to be included in the next DCESS II update. This was mentioned, albeit rather indirectly, in the next-to-last paragraph in section 4 ("... . In another example, planned incorporation of methane and nitrogen cycles into DCESS II will significantly improve its ability to deal realistically with deep-time global warming events associated with massive carbon inputs to the Earth System. ..."). Furthermore, in that paragraph it was pointed out that "... . Groundwork for this step has already been done in as much as these global biogeochemical cycles were incorporated into the DCESS I model (Shaffer et al., 2017). ...". Indeed in that publication, methane, nitrogen and sulfur cycles were included in the DCESS I model to deal with nitrogen fixation, denitrification and sulfate reduction for suboxic/anoxic ocean conditions. In order to clarify this, we rewrote the central part of this paragraph to read:

*"... . In another example, planned incorporation of methane, nitrogen and sulfur cycles into the next DCESS II update will significantly improve its ability to deal realistically with deep-time global warming events associated with massive carbon inputs to the Earth System. Under such conditions, suboxic/anoxic ocean conditions may arise leading, for example, to denitrification and sulfate reduction that must be addressed in such an update. Groundwork for this step has already been done in as much as these global biogeochemical cycles and suboxic/anoxic processes have been incorporated into the DCESS I model (Shaffer et al., 2017). ..."*

It is impressive that the distribution of ocean circulation tracers and biogeochemistry is in very good agreement with modern observations, but this is a separate issue from the question of whether correct projections can be made when the system deviates from the steady state. The dynamic behavior of the model is also examined in section 3.2, but it remains unclear why the results were concluded to be "reasonable" (Line 30), since there is no explicit comparison with the results of more complicated models.

We concede that the use of "reasonable" in line 30 is somewhat subjective. In response, we rewrote that sentence of the abstract to read:

*"... . Changes in ocean circulation and in the global carbon cycle found in these experiments are in line with  results from much more complex models. ..."*

We note that we did compare with results of more complex models in section 3.2: In the last sentence of the first paragraph in section 3.2.1 we write: "Insensitivity of North Atlantic AMOC intensity as well as intermediate-depth Southern Hemisphere recirculation for increased westerlies forcing has also been reported in other modelled experiments (Jochum and Eden, 2015; Rahmstorf and England, 1997)." However, the referee makes valid point that clearly more comparisons with published work are called for. To this end we made the following additions:

In the second paragraph of section 3.2.1, we include:

"… . *Such a Southern Hemisphere westerly winds intensity-atmospheric $CO_2$ relationship has also been reported in another modelled experiment (d'Orgeville et al., 2010). …*"

And

"… . *Finally, our Exp1a results are in line with the importance of Southern Hemisphere westerly winds on deglacial atmospheric $CO_2$ rise as found in models and proxy-data reconstructions (Mayr et al., 2013; Menviel et al., 2018).*"

Furthermore, in the first paragraph of section 3.2.2, we include

"… . *Such effect on AABW production rates due to Antarctic meltwater on Southern Ocean shelf has been observed in similar Antarctic-meltwater experiments using more complex models (Fogwill et al., 2015; Silvano et al., 2018; Li et al., 2023). …*"

And

"… . *Such behavior has been reported for numerous model experiments including a multi-model ensemble of CMIP6 model experiments (Park and Latif, 2019; Mackie et al., 2020; Chen et al., 2023). …*"

And

"… . This suggests ageing of the abyssal ocean, also reported in Li et al., (2023). …"

In the second paragraph of section 3.2.2, we include:

"… . *This demonstrates the model's ability to capture the $\delta^{18}O_w$ signal from Antarctic ice-sheet melting, shown to be a better indicator of such melting than ocean salinity changes (Kim and Timmermann, 2024). …*"

Finally, we complement our discussion from these experiments (Exp1 and Exp2) at the end of the third paragraph of Section 4 by including:

"… . *Our Exp1 supports the role of the Southern Hemisphere westerly winds in modulating past glacial-interglacial, atmospheric carbon dioxide variations, while our Exp2 points toward the impact*

*that future Antarctic ice sheet melting would have on the formation of Antarctic Bottom water and on the deep global ocean in general. ..."*

Detailed comments:

Line 18 and Figure 1 caption: "ocean boxes" –ocean regions (or sectors)?

We changed "ocean boxes" to "ocean sectors" throughout the manuscript.

Line 30: "Changes in ocean circulation and in the global carbon cycles found in these experiments are reasonable and agree with results for much more complex models." –It is unclear, at least for this reviewer, how the validity of the results obtained was assessed, because no explicit comparison with the behavior of more complex models was given.

This is discussed and addressed above including a new formulation of this line

Line 49—55: Perhaps a comment could be added here about the motivation for further model development. I feel it is unclear what the limitations of the original model were and what problems will be approached by improving the geometry and physical/biogeochemical processes in this study.

This was dealt with above under the first Major Comment.

Line 96: It might be nice to explain why three vegetation types need to be considered.

The referee is correct that we need to better explain and motivate the use of the three vegetation types but the best place to do that is at the beginning of section 2.6 on the land biosphere module. Accordingly, the first paragraph of that section now reads:

*"Eichinger et al., (2017) defined three different, dynamically varying vegetation zones to extend and improve the original DCESS I, one-zone land biosphere module. The vegetation zones - a grassland/desert zone bordered equatorward and poleward by tropical forest and extra tropical forest zones, respectively - were formulated by emulating the behavior of a complex land biosphere model (Gerber et al., 2004). With this approach, latitudinal boundaries of the zones could be defined as functions of global mean temperature alone encompassing implicit dependency on precipitation. In this way, the very different carbon distributions between, say, above-ground biomass and soil for each zone and the responses of these carbon reservoirs to changing climate and atmospheric $CO_2$ could be addressed. For example, while most of the carbon in tropical forests is found in above-ground, by far most of the carbon in extratropical forests is in the soil (Chapin III et al., 2011). With this new three-zone module, the size and timing of carbon exchanges between atmosphere and land were represented much more realistically in cooling and warming experiments than with the original, one- zone module (Eichinger et al., 2017). Furthermore, our three-zone approach allows for changing biosphere modulation of radiative forcing since albedo is higher for grasslands/deserts than for forests (see albedo formulations in section 2.1 above).*

*Here we use the same approach but now expanded to two hemispheres. The vegetation zones are ...”*

Line 480: Estimating decomposition/dissolution rates based on e-folding length is a simple and useful approach, but in practice, the decomposition/dissolution rate at each water depth is influenced by several environmental factors, such as sinking velocity and the degree of saturation of seawater with respect to carbonates.

There are certainly a number of ways to address the problem of ocean decomposition/dissolution rates. The formulation we use is based on global observational data (see Shaffer et al., 1999), and a such captures implicitly such ambient factors you mention. As simple and useful, this e-folding length-type model approach has been extensively used (Ridgwell and Hargreaves, 2007; Komar and Zeebe, 2021; Ozaki et al., 2022). Furthermore, now we have extended our original DCESS I model approach to consider local temperature-dependence on organic matter remineralization rates. This goes a long way to capture the influence on these rates from spatial and temporal changes in ocean environmental conditions (Berelson, 2001; Lutz et al., 2002; Marsay et al., 2015; Laufkotter et al., 2017).

Eq. (43): The vegetation type would affect the rate of chemical weathering. The rationale behind this simplification is left unstated/unjustified in the current manuscript.

We prefer to maintain our simplified dependence of weathering rates on temperature only. We feel that incorporating explicit variations in chemical weathering rates due to vegetation type cover would add extra, highly-tunable complexity and would be beyond the scope and balance of our simplified model. In as much as we scaled weathering rates and associated river inputs to the ocean to balance calculated ocean sediment burial rates in our pre-industrial calibration, present day weathering of course includes implicitly the effects of vegetation on Earth. And an increase of weathering rates with increasing temperature would be consistent with increasing vegetation coverage predicted for such conditions for our land biosphere module.

This is now discussed shortly in the second paragraph of section 2.7 to read:

*“... . Vegetation affects weathering rates by modifying surface pH through production of $CO_2$ or organic acids or by altering the physical properties of soil such as erosion of exposed mineral areas and by water cycling content (Drever, 1994; Berner, 1995). Our model does not explicitly include these and other factors like a direct dependency of atmospheric $pCO_2$ levels (Krissansen-Totton and Catling, 2017). Such factors would add extra tuneable complexity and be beyond the scope and balance of our simplified model. ...”*

Line 705: "0.2095 $\mu$atm" --0.2095 atm?

You are right, the correct units are 0.2095 atm. This has been corrected in the text.


condition, while I think it is not necessarily required if a comparison of the sensitivity tests that have already done in the manuscript with the results of previous studies conducted using sophisticated ocean models is enriched sufficiently in the course of revision.

YW is correct that more comparisons with published work using more sophisticated models are needed to help validate the performance of our new, simplified model. This comment is similar to that voiced by referee 1. In future work with DCESS II on projecting future changes in the Earth System we plan to make detailed comparisons with historical changes as further model verification.

For the present to address the above point, we made the following changes/additions as also noted in our detailed response to referee 1:

We rewrote the last sentence in the abstract to read:

*"... . Changes in ocean circulation and in the global carbon cycle found in these experiments are in line with results from much more complex models. ..."*

In the second paragraph of section 3.2.1, we include:

*"... . Such a Southern Hemisphere westerly winds intensity-atmospheric $CO_2$ relationship has also been reported in another modelled experiment (d'Orgeville et al., 2010). ..."*

And

*"... . Finally, our Exp1a results are in line with the importance of Southern Hemisphere westerly winds on deglacial atmospheric $CO_2$ rise as found in models and proxy-data reconstructions (Mayr et al., 2013; Menviel et al., 2018)."*

Furthermore, in the first paragraph of section 3.2.2, we include

*"... . Such effect on AABW production rates due to Antarctic meltwater on Southern Ocean shelf has been observed in similar Antarctic-meltwater experiments using more complex models (Fogwill et al., 2015; Silvano et al., 2018; Li et al., 2023). ..."*

And

*"... . Such behavior has been reported for numerous model experiments including a multi-model ensemble of CMIP6 model experiments (Park and Latif, 2019; Mackie et al., 2020; Chen et al., 2023). ..."*

And

"... . This suggests ageing of the abyssal ocean, also reported in Li et al., (2023). ..."

In the second paragraph of section 3.2.2, we include:

*"... . This demonstrates the model's ability to capture the $\delta^{18}O_w$ signal from Antarctic ice-sheet melting, shown to be a better indicator of such melting than ocean salinity changes (Kim and Timmermann, 2024). ..."*

Finally, we complement our discussion from these experiments (Exp1 and Exp2) at the end of the third paragraph of Section 4 by including:

*"... . Our Exp1 supports the role of the Southern Hemisphere westerly winds in modulating past glacial-interglacial, atmospheric carbon dioxide variations, while our Exp2 points toward the impact that future Antarctic ice sheet melting would have on the formation of Antarctic Bottom water and on the deep global ocean in general. ..."*

**2. Configuration of the lithosphere module and the treatment of the budgets of atmospheric species**

Reading through the manuscript, the configurations of the lithosphere module and the budget of the atmospheric species, especially oxygen, in the model seem to be odd to me, partly because the performance of this module has not been compared with the global biogeochemical cycle models. First, I think that a consideration of the global sulfur cycle would be required for reproducing the steady state of the atmospheric oxygen level in a proper way, as done in many previous global carbon-sulfur-oxygen models.

YW is correct that if the model were to be used with the intention of addressing slow, multi-million changes in the "steady state" of atmospheric $O_2$ then the addition of a global sulfur cycle would be needed and this cycle will surely be included in future model versions. The DCESS II treatment of atmospheric $O_2$ in section 2.2 (now described in more detail; see below) is essentially identical to that used in DCESS I (Shaffer et al., 2008). In that publication, a long 1.5-million-year simulation was carried out to investigate the consequences of doubling lithosphere outgassing of $CO_2$. By the end of the simulation, a new climate steady state has nearly been reached with global mean temperature of 24.5 °C and an atmospheric $p$CO$_2$ of 2636 ppm. By then, atmospheric $O_2$ had increased slightly (due to enhanced organic carbon burial) from an initial value of 0.2095 atm to 0.2112 atm but continued increasing. This illustrates that large, natural changes in atmospheric $O_2$ can only be achieved on multi-million year time scales for which as stated in the publication "A proper treatment of the coupled carbon, nutrient, oxygen and climate system over such long time scales is beyond the scope of the present model and would require, for example, a treatment of sulfur cycling (Berner, 2006)." On the other hand, the model treatment of atmospheric $O_2$ in DCESS I has proven to be quite useful on shorter time period in the context of present and future anthropogenic change (Shaffer et al., 2009; Keeling et al., 2021) and surely will continue to be so with DCESS II.

Second, the formulation of the weathering rates seems to be different from the formulation that has been widely used using the conventional global carbon cycle, which makes it unclear whether their formulation can be applied to various conditions in the geologic time.

We address this point in detail below showing that our results agree with those of more complex approached and arguing that our simple formulation is consist with approaches used in other model components and avoids extra tunable parameters.

Third, I concern the treatment of the atmospheric budget of oxygen, methane, and CO2. When considering both the fast and slow processes of the atmospheric (photo)chemical reactions, one should pay careful attention to the budget of atmospheric species in the atmosphere and should explain them comprehensively in the manuscript. The details of these concerns are found in the following line-by-line comments.

We address this point in detail below.

**Line-by-line specific comments (The capital L represents line number):**

L65: It is unclear what "seasonal cycles" means. Is it a seasonal cycle of climate and/or atmospheric pCO2?

To avoid any confusion at this stage we rewrote this line to read:

"... . *It is an enhancement and extension of the original DCESS I model (Shaffer et al., 2008) and includes, for example, much improved horizontal and time resolution as well as simplified ocean dynamics. ...*"

L100: Why does air-sea exchange affect the energy balance of the atmosphere? If the intended meaning is that the atmospheric pCO2 changes owing to the air-sea exchange, it should be specified clearly.

No, the exchange of heat with the ocean is referred to here. To avoid confusion, we reformulated the first line in section 2.1 to read:

"*We use a simple, zonally integrated energy balance model for the near surface atmospheric temperature, $T_a$ (°C), forced by seasonally varying insolation, heat exchange with the ocean, and meridional heat transport. ...*"

L110: ... and the ocean surface, respectively, ...

This correction has been made in the text.

L115: Does this Ta(Φ) correspond to the temperature at any given latitude, different from the temperature of each atmospheric box? It would be helpful if more comprehensive explanation is given.

To clarify this, we reformulated the line before equation (2) to read:

*"... . At each point in time, we construct atmospheric temperature as a continuous function of latitude, $T_a(\phi)$, using a fourth-order Legendre polynomial in sine of latitude $\phi$,"*

L166–167: The application limit of the formulations of Byrne and Goldblatt should be specified (i.e., 200–10000 ppmv for CO2, 0.1–100 ppmv for CH4 and N2O if I understand correctly). Also, it is worth noting that the formulation considers the overlap of the absorption by N2O with CH4 and CO2.

We now include the application limits and overlap information is the last paragraph of section 2.1 to read:

*"... where expressions for $A_{CO_2}$, $A_{CH_4}$ and $A_{N_2O}$, are taken from Byrne and Goldblatt (2014) that are valid for atmospheric concentrations in the range of 200 – 10,000 ppm for $CO_2$ and 0.1 – 100 ppm for $CH_4$ and $N_2O$. Overlap of the absorption by $N_2O$ with $CO_2$ and CH4 is included in these formulations. ..."*

L168–169: Could these preindustrial values be justified by comparing with the observation values from ice cores or else?

In response we now take the corresponding line in the last paragraph of section 2.1 to read:

For clarity, lines 168-169 will read:

*"... . We take the year 1765 as our pre-industrial (PI) baseline where values of $pCO_{2,PI}$, $pCH_{4,PI}$ and $pN_2O_{PI}$ are 280, 0.72 and 0.27 ppm, respectively, as indicated by ice-core data (IPCC, 2021 and citations therein). ..."*

L226: Please specify how much the climate should be warm for "extremely warm climates" that requires the Schmidt number formulation.

This has now been specified at the end of the second paragraph of section 2.2 to read:

*"... . For extremely warm climates experiments (annual mean sea-surface temperatures over 30 °C) we use the Schmidt number formulation from Gröger and Mikolajewicz (2011) who demonstrated that the Wanninkhof (1992) formulation underestimates Sc in such conditions."*

L275–295: These sentences would correspond to the explanation of the terms $\Psi_I(\chi)$, but I think it is clearer to formulate these terms in equations as done for $\Psi_S(\chi)$ and $\Psi_T(\chi)$.

We prefer to retain this as is in order to maintain a similar structure as in DCESS I model description (Shaffer et al., 2008).

L275: Should "(see Sect. 2.5)" be "(see Sect. 2.6)"?

This has been corrected.

L277: Is $\lambda_{CH4}$ $pCH_4$ in the sentence an equation? If so, isn't this equation necessary to be normalized by the preindustrial pCH4 value?

We now in this paragraph include a more complete discussion of how the atmospheric lifetime of methane and its associated atmospheric sink are calculated and what consequences this sink has for atmospheric $O_2$ and $CO_2$:

*"For methane there is production within the land biosphere (see Sect. 2.6) and consumption associated with OH radicals in the troposphere. The latter leads to the net consumption of 2 $O_2$ molecules and production of 1 $CO_2$ molecule for every $CH_4$ molecule consumed. Since this reaction depletes the concentration of these radicals, atmospheric lifetime of $CH_4$ grows as methane concentration rises. We include this effect in the model by taking the atmospheric methane sink to be $\lambda_{CH_4}$ $pCH_4$, with $\lambda_{CH_4} = v_a/\tau_{CH_4}$ where $\tau_{CH_4}$ is the (variable) atmospheric lifetime of methane. This lifetime is determined by fitting a function to the results from several modelling studies that consider a wide range of $pCH_4$ values. This fit yields a pre-industrial lifetime of 9.5 years, increasing for example to 10.8 and 15.1 years for 2 and 10 times the pre-industrial methane level, respectively (see Shaffer et al., 2017 for details)."*

L289: The budget of oxygen in the atmosphere should be explained in more detail using equations. I have several concerns regarding the treatment of oxygen in the model. My first concern regarding this is the representation of the atmospheric pO2 budget by the oxidation of methane and reactions with OH radicals. I agree that the net (photo)chemical reactions would work as a net sink of oxygen during the oxidation of methane in the atmosphere because part of OH radicals originate from oxygen via many chemical reactions. As a result, the net reaction is represented as follows:

CH4 + 2O2 –> CO2 + 2H2O

A similar approach can be seen in a previous study (Goldblatt et al., 2006 Nature). The produced CO2 in the above net chemical reaction compensates the net consumption of CO2 when producing methane. Is this treated in the source and sink terms of O2 and CO2 properly? In addition, I'm not sure why a reaction of oxygen with OH radicals can be a net sink of oxygen. Obviously, oxygen is consumed and produced by many fast (photo)chemical reactions. An example is the Chapman mechanism that forms ozone in the stratosphere. However, the key for understanding the net budget of oxygen in the atmosphere is the net reaction as in the case of the methane oxidation mentioned above. For the case of the reaction with OH radicals, however, I do not come up with the mechanism that works as the net sink of oxygen in the atmosphere. The authors should explain the configurations and assumptions regarding the budgets of atmospheric chemical species in the model clearly.

We agree with YW that the description of our treatment of the atmosphere oxygen budget of oxygen in the atmosphere could be improved. We feel however that this does not require an introduction of additional equations but rather, as for carbon dioxide and methane, the use of the general equation

(12) and a more detailed description of source/sink terms. YW states above that "The produced CO2 in the above net chemical reaction compensates the net consumption of CO2 when producing methane. Is this treated in the source and sink terms of O2 and CO2 properly? In addition, I'm not sure why a reaction of oxygen with OH radicals can be a net sink of oxygen". However, there is no net consumption of $CO_2$ in the production of methane for the processes considered in the model. Rather, as detailed in section 2.6, the only source of methane to the atmosphere is by the action of bacteria in remineralizing soil organic matter. Therefore, the above reaction is also a net source of $CO_2$ and a net sink of $O_2$ as applied in the model (see section 2.2).

In accordance with the above, we improved the second-to-last paragraph in section 2.2 to read:

"*For oxygen, there is consumption associated with oxidation of atmospheric methane and reaction with OH radicals (see above), and sink (source) from organic matter remineralization (photosynthesis) on land. Since methane is the end product of some of the remineralization on land (see section 2.6), the land biosphere is a net $O_2$ source in a steady state. Furthermore, there are long-term atmospheric oxygen sinks due to weathering of organic carbon in rocks and oxidation of reduced carbon emitted in lithosphere outgassing. A long-term, quasi-steady state of $pO_2$ is achieved in the model when these latter sinks balance net $O_2$ outgassing from the ocean from less $O_2$ consumption than production there due to burial of organic matter in the model ocean sediments. However, for multi-million-year time scales, the global sulfur cycle would need to be included in the model to achieve "true" $pO_2$ steady states (Berner, 2006). Additional sinks (sources) of atmospheric $O_2$ associated with recent land use change and with burning of fossil fuels may be included in the model as needed.*"

Second, the long-term steady state of the atmospheric pO2 should also be affected critically by the oxidation of continental sulfide minerals (i.e., pyrite) and the deposition of pyrite from the ocean. Many previous studies of the oxygen biogeochemical cycle in geologic time consider the global sulfur cycle (Berner et al., 2000 Science; Berner, 2001 GCA; 2006 GCA; Bergman et al., 2004 Am. J. Sci.; Lenton et al., 2018 Earth-Sci. Rev.; Krause et al., 2018 Nature Commun.; Ozaki and Reinhard, 2021 Nature Geosci.; Ozaki et al., 2022 GMD). I think the consideration of the global sulfur cycle is required to reproduce the long-term steady state of the present-day atmospheric pO2 (~0.21 atm). I think it can be treated easily in the framework of DCESS II model by considering the global sulfur budget in the model as in the above previous studies. Alternatively, I think that not treating the atmospheric oxygen in the model would also be okay because it is not the primary focus of the model.

This was addressed above expressing agreement that indeed a global sulfur cycle would be required for multi-million-year simulations of atmospheric $O_2$. Likewise, this was noted in the revised, second-to-last paragraph in section 2.2. It was also demonstrated with published examples that the treatment of atmospheric $O_2$ in the original DCESS I models has been very fruitful for shorter time scales. Since DCESS II adopts the DCESS I treatment this will likely be the case for future

applications of the new model. No, the DCESS II treatment of atmospheric $O_2$ should not be abandoned.

L569–575: The configuration of the land ecosystem should be explained in more detail. Specifically, the formulation of the meridional limits for each vegetation zone would be, at most, very rough. For example, the combined grasslands-savanna-desserts zone should, in reality, include the dessert regions, the grassland of C4 plants in tropical and subtropical regions, and the grassland of C3 plants in extratropical regions including the tundra region near the Arctic. Although this rough treatment may be acceptable considering the simplicity of the DCESS II model, it is unclear to me what is the purpose of the consideration of the three vegetation zones in such a simplified model. Given that the relationship between atmospheric pCO2 and the land NPP in equation (36) is common for three vegetation zone (except for the small difference in $NPP_{PI}$), I'd rather prefer a uniform distribution of vegetation in such a simplified land model (such as, Lenton and Huntingford, 2003 GCB). I guess that the authors introduced the shift of vegetation zones because it improves the estimation of the reproducibility of the land NPP and/or the carbon distributions in litter and soil reservoirs. If so, this treatment would still be acceptable, but the limitation of the configuration of land vegetation should be clearly explained in the text.

The three-zone land biosphere module was originally developed to replace the one-zone module of the DCESS I model (Eichinger et al., 2017). We feel that it strikes the right balance for use in the upgraded but still simplified DCESS II model. However, YW is correct that more explanation is needed in the manuscript to described the module and its advantages. To this end we revised and extended the first paragraph of section 2.6 to read:

*"Eichinger et al., (2017) defined three different, dynamically varying vegetation zones to extend and improve the original DCESS I, one-zone land biosphere module. The vegetation zones - a grassland/desert zone bordered equatorward and poleward by tropical forest and extra tropical forest zones, respectively - were formulated by emulating the behavior of a complex land biosphere model (Gerber et al., 2004). With this approach, latitudinal boundaries of the zones could be defined as functions of global mean temperature alone encompassing implicit dependency on precipitation. In this way, the very different carbon distributions between, say, above-ground biomass and soil for each zone and the responses of these carbon reservoirs to changing climate and atmospheric $CO_2$ could be addressed. For example, while most of the carbon in tropical forests is found in above-ground, by far most of the carbon in extratropical forests is in the soil (Chapin III et al., 2011). With this new three-zone module, the size and timing of carbon exchanges between atmosphere and land were represented much more realistically in cooling and warming experiments than with the original, one-zone module (Eichinger et al., 2017). Furthermore, our three-zone approach allows for changing biosphere modulation of radiative forcing since albedo is higher for grasslands/deserts than for forests (see albedo formulations in section 2.1 above).*

*Here we use the same approach but now expanded to two hemispheres. The vegetation zones are ...*"

L569: Does the change in surface vegetation feedback to the surface albedo? Regardless, this should be mentioned in this section.

See above paragraph. For example, as the Earth cools from present day, both forested areas (tropical and extra tropical) contract while the non-forested area (grassland-savanna-dessert) expands slightly, in part due to drier conditions (Gerber et al., 2024). This leads to higher albedo and positive feedback on the cooling. This effect is included into the model through the factor $\alpha_0$ in the surface albedo formulation as already described in Section 2.1.

L595: There should be an upper limit of the atmospheric pCO2 for the condition in which the relationship between atmospheric pCO2 and the land NPP holds. It may be worth mentioning this because this model may be applied to a geologic event with extremely high atmospheric pCO2 (for example, > 2,000 ppm), such as the eruption of the Ontong Java Plateau and Caribbean Plateau during the Cretaceous, and/or a distant future of a severe global warming scenario.

Thanks to YW for pointing out the need to address extreme situations in the text. There is an upper limit in our three-zone approach and it is associated with the hemispheric annual mean temperature deviation from its pre-industrial value ($\delta Ta$). The value of $\delta Ta$ cannot be greater than 10°C, i.e., the modelled hemispheric annual mean temperature must be less than 25 °C that would correspond to an atmospheric $pCO_2$ of around 1000-1500 ppm. This is because our fitted polynomials approach for meridional limits of vegetation zones is based on results of Gerber et al., (2004) who run their experiments only in the range of $\delta Ta$ -10 °C to 10 °C. This limitation and its implications are now stated in the text at the end of paragraph 2 of section 2.6:

*"…. We note that the above approach is only strictly valid for -10°C < $\delta T_a$ < 10 °C, the range considered in the original experiments of Gerber et al. (2004). For warming this corresponds to annual mean temperatures less than about 25 °C and corresponding atmospheric $pCO_2$ levels less than about 1000-1500 ppm. For more extreme warming situations in the distant past or future a re-evaluation of our land biosphere module would be necessary for use in model simulations."*

L605–614: The explanation about the allocation of carbon to leaves and wood (the numbers in equations 37 and 38) is found only in the previous description paper (Shaffer et al., 2008). It would be helpful if the assumptions regarding this are explained at least briefly in the present study.

Those factors are based on Siegenthaler and Oeschger's (1987) model and are adopted in Shaffer et al., (2008) as well as in Eichinger et al., (2017). As recommended by YW, we now add these allocation assumptions at the beginning of paragraph 3 in section 2.6:

*"With the descriptions above and the assumptions that assuming that NPP is distributed between leaves and wood in the fixed ratio 35:25, all leaf loss goes to litter, wood loss is divided between litter and soil in the fixed ratio 20:5, and litter loss is divided between the atmosphere (as $CO_2$) and the soil in the fixed ratio 45:10 (Siegenthaler and Oeschger, 1987; Shaffer et al., 2008), the conservation equations …"*

L640: Usually, the conventional global carbon cycle models consider both temperature- and CO2-dependency on silicate and carbonate weathering rates (e.g., Berner, 1991 Am. J. Sci.; Tajika, 2003 EPSL; Bergman et al., 2004 Am. J. Sci.; Berner, 2006 GCA; Krissansen-Totton and Catling, 2017 Nature Commun.; Krissansen-Totton et al., 2018 PNAS). I believe that the temperature dependency using Q10 function (Q10 = 2) would also reproduces a similar dependency to the conventional models, but this would limit the applicability of the model when the relationship between the atmospheric pCO2 and surface temperature is different from the present-day condition (e.g., under very high atmospheric pCO2, different solar luminosity, different continental areas, lack of land plants, etc.). I recommend using the conventional dependency on continental weathering rates, but if not, I recommend enriching the explanation about the validation of the formulation of the weathering rates comparing with the previous conventional global carbon cycle models and mentioning the known limitations of this treatment.

More complex weathering approaches introduce several poorly constrained factors such as the weatherability factor which encompasses changes in land area due to sea-level variations, changes in lithology, relief, vegetation and palaeogeography. Our simplified approach is designed to avoid a plethora of tunable parameters, whereby we anchor weathering variations to the simple Q10 approach used to describe temperature-dependent rates throughout all our model modules. Furthermore, as noted above, use of our land biosphere module is only at present justified for annual mean temperatures less than about 25 °C and corresponding atmospheric $pCO_2$ levels less than about 1000-1500 ppm. For these ranges the temperature dependency using Q10 function (Q10 = 2) would probably also reproduce a similar dependency to the conventional models with additional $pCO_2$ dependency. For more extreme warming situations, as for our land biosphere module, a re-evaluation of our weathering treatment may be necessary for use in model simulations.

All this is now discussed shortly in the second paragraph of section 2.7 to read:

"... . *Vegetation affects weathering rates by modifying surface pH through production of $CO_2$ or organic acids or by altering the physical properties of soil such as erosion of exposed mineral areas and by water cycling content (Drever, 1994; Berner, 1995). Our model does not explicitly include these and other factors like a direct dependency of atmospheric $pCO_2$ levels (Krissansen-Totton and Catling, 2017). Such factors would add extra tuneable complexity and be beyond the scope and balance of our simplified model. ...*"

Also, we follow YW´s recommendation to complement and enrich our lithosphere module results in section 3.1.7. We do this below in the context of the specific comments.

L683–690: This section seems to correspond to Model description, not to Model solution, calibration and validation.

Thank you for your suggestion, however, we prefer to retain this section as is to maintain a similar structure as in the DCESS I model description (Shaffer et al., 2008).

L695: References for a climate sensitivity of 3 ˚C may be added (e.g., Meehl et al., 2020 Sci. Adv.; Zelinka et al., 2020 GRL).

*In response we included additional references in the first paragraph of section 3.1.2 to now read:*

*"... annual mean atmospheric temperature of 15 °C and a climate sensitivity of 3 °C per doubling $CO_2$ as indicated by several lines of evidence and model estimates (Meehl et al., 2020; Zelinka et al., 2020; IPCC, 2021), a poleward transport ..."*

L700: Is the function f(I) and the parameters in equation (36) (i.e., f0, a, b, and c) different between atmospheric sectors and/or vegetation zones? This should be clarified near equation (36).

*There is one f(I) function (with their respective parameters) for each vegetation type (i.e. a total of three f(I) functions). This is now included in the third paragraph of section 2.6 to read:*

*"... each zone equal to one. Note that each vegetation type has its own function $f(I)$ with its respective parameters. With this formulation ..."*

L702: The amplitude of the seasonal variation of the atmospheric pCO2 varies spatially as can be seen in atmospheric pCO2 reanalysis data (e.g., Maki et al., 2010 Tellus B), which is especially large near the boreal forest region. Is the model tuned to reproduce the amplitude in each atmospheric sector or to reproduce the global mean amplitude? I think the detail of the calibration of the atmospheric pCO2 and the reference of the observation should be provided.

*There is no such explicit calibration for atmospheric $pCO_2$. The amplitude of seasonal variation of atmospheric $pCO_2$ is largely set by the annual cycle of land biosphere NPP, modulated by the set of functions $f(I)$ fitted to reproduce this cycle for each vegetation type in the model.*

*This is expressed in the first paragraph of section 3.1.2 to read:*

*"... and we adjust free parameters of the set of functions $f(I)$ in Eq. (36) to give observed annual mean NPP values for each vegetation zone as well as their annual cycle amplitudes from observations. This largely sets the modelled annual cycle of atmospheric carbon dioxide of each atmospheric box. ..."*

L727: "atom m-2 s-1" –> "atom m-2 s-1 respectively"

*This correction has been made.*

L795: How does the extensive sea ice increase the deep ocean temperature in the Arctic?

*The model assumes no air-sea heat exchange in the presence of sea ice. As the model annual mean sea-ice line position is at ~66 °N, there is almost no ocean to the atmosphere flux in the Arctic Ocean*

sector with its equatorward limit at 65°N. Our zonally-mean, model approach is not able to capture zonally asymmetries that are present in the Arctic Ocean like coastal polynyas and local deep water formation sites. In this situation, our model Arctic is warmed by ocean heat transport from the south.

L913: It may be worth noting that the organic carbon burial flux of 0.11 GtC yr–1 (~9 Tmol yr–1) is broadly consistent with the values in conventional global carbon cycle models (e.g., Berner, 1991 Am. J. Sci.; Bergman et al., 2004 Am. J. Sci.).

We agree and now we compare our model results with both data-based estimates and model estimations for organic carbon as well as for calcite carbon. For organic carbon, the first paragraph in section 3.1.6 now reads:

*"In the model pre-industrial steady state, 0.11 Gt C yr$^{-1}$ of organic carbon is buried in ocean sediments. This represents 12 % of total organic carbon falling on sediment surface; the remaining 88 % is remineralized to DIC that returns to the water column. This modelled value is well within the range of data-based estimations (Berner, 1982; Hedges and Keil, 1995; Cartapanis et al., 2018). However, the higher ends of some of these estimates may include contributions from estuaries and delta environments that our model does not consider. Other modelled results are in the range of 0.02 – 0.09 Gt C yr$^{-1}$ (Berner 1991; Munhoven, 2007; Willeit et al., 2023). More than 90 % of our modelled burial takes place at water depths shallower than 1000 m such that 0.011 Gt C yr$^{-1}$ is buried below this depth, in agreement with deep sea burial estimates (Hayes et al., 2021). The highest modelled burial rates for organic carbon per ocean sector are found in the tropical Pacific Ocean and the North Atlantic Ocean (35° N-65° N) sectors with over 40 % and 27 % of the total organic carbon burial, respectively. The global inventory of bioturbated layer (BL) organic carbon is 122 Gt C, somewhat higher than found in the pre-industrial, DCESS I model simulation (92 Gt C)."*

For calcite carbon, the first part of the second paragraph of section 3.1.6 now reads:

*"For calcite, the global annual mean burial rate is 0.21 Gt C yr$^{-1}$ where more than 57 % takes place in the tropical Pacific sectors and 23 % is buried at the combined north-tropical and mid-latitude Atlantic Ocean sectors. Our global result compares well with data-based estimations and model results in the range of 0.13 – 0.45 Gt C yr$^{-1}$ (Milliman and Droxler, 1995; Cartapanis et al., 2018; Hayes et al., 2021; Willeit et al., 2023). Of the total annual mean burial rate of $CaCO_3$ in the Pacific and Atlantic Ocean (0.14 and 0.07 Gt C yr$^{-1}$ respectively) 49 % and 45 % of calcite is buried at depths greater than 1000 m in these basins, respectively. Other estimates also show such a nearly half-to-half, shelf-slope vs deep ocean division (Milliman, 1993). ..."*

L955–968: The values in Table 5 seems to be okay to me, but I think they should be compared with observational and/or theoretical estimates following the previous studies using the global carbon cycle models mentioned in the above comments.

We agree and now we compare with other estimate such that the last part of the first paragraph in section 3.1.7 now reads:

*"... in good agreement with to modern-day data-based estimates (IPCC, 2021). Results for carbonate and silicate weathering rates are similar to those found with the DCESS I model, 0.109 and 0.092 Gt C yr$^{-1}$, respectively (Shaffer et al., 2008). Our carbonate weathering rate is well into the range of other observational and model estimates, 0.088 – 0.241 Gt C yr$^{-1}$, but our silicate weathering rates are somewhat lower than other corresponding estimates, 0.122 – 0.236 Gt C yr$^{-1}$ (Gaillardet et al., 1999; Ludwig et al., 1999; Munhoven, 2002; Hartmann et al., 2009; Lenton et al., 2018; Willeit et al., 2023). Phosphorus weathering rates are higher than those found in the DCESS I model, 1.663×10$^3$ mol P s$^{-1}$ but are in better agreement with observational estimates, 2.019 – 3.071×10$^3$ mol P s$^{-1}$ (Filippelli, 2002; Paytan and Mclaughlin, 2007). Organic carbon weathering rates agree well with recent estimations of 0.068 Gt C yr$^{-1}$ (Zondervan et al., 2023). Based on these results, although simple, our continental weathering-climate relationship gives reasonable results when compared against models having more complicated relationships."*

L986: It may be worth mentioning here that the strengthened clockwise recirculation cell at 40–55˚S is also consistent with Rahmstorf and England (1997), although it has already been stated briefly at L991–993.

We prefer to retain our original text.

L1003–1004: Should "Exp1 and Exp2" be "Exp1a (Exp1b)"?

This has been corrected.